



# Experimental validation of a short-term damping estimation method for wind turbines in nonstationary operating conditions

Kristian Ladefoged Ebbehøj[1,2], Philippe Jacques Couturier[3], Lars Morten Sørensen[2], and Jon Juel Thomsen[2]

[1]Department of Civil and of Mechanical Engineering, Technical University of Denmark, Denmark
[2]Siemens Gamesa Renewable Energy A/S, Denmark
[3]Siemens Gamesa Renewable Energy Inc., CO, USA

**Correspondence:** Kristian Ladefoged Ebbehøj (kleb@dtu.dk)

**Abstract.** Modal properties and especially damping of operational wind turbines can vary over short time periods as a consequence of environmental and operational variability. This study seeks to experimentally test and validate a recently proposed method for short-term damping and natural frequency estimation of structures under influence of varying environmental and operational conditions from measured vibration responses. The method is based on Gaussian Process Time-dependent Auto-
Regressive Moving average (GP-TARMA) modelling, and is tested via two applications: a laboratory three-storey shear frame structure with controllable, time-varying damping, and a flutter test of a full-scale 7 MW wind turbine prototype, in which two edgewise modes become unstable. Damping estimates for the shear frame compare well with estimates obtained with Stochastic Subspace Identification (SSI) and standard impact hammer tests. The efficacy of the GP-TARMA approach for short-term damping estimation is illustrated through comparison to short-term SSI estimates. For the full-scale flutter test GP-TARMA
model residuals imply that the model cannot be expected to be entirely accurate, but the damping estimates are physically meaningful, and compare well with a previous study. The study shows that the GP-TARMA approach is an effective method for short-term damping estimation from vibration response measurements, provided enough training data and a representative model structure.

## 1 Introduction

A novel Operational Modal Analysis (OMA) method (Ebbehøj et al., 2023) for short-term modal damping estimation for structures under influence of varying Environmental and Operational Conditions (EOCs), such as wind turbines in operation, is tested with a controlled laboratory experiment and a wind turbine flutter test.

The dynamic properties of wind turbines (i.e., natural frequencies, damping ratios, and mode shapes) can be sensitive to changing EOCs (Avendaño-Valencia et al., 2017; Bogoevska et al., 2017). Aeroelasticity, active control, material properties,
and nonlinear damping mechanisms (e.g., friction) are examples of phenomena and factors which can cause EOC variability (Hansen et al., 2006b; Wang et al., 2022; Chen and Duffour, 2018). EOC variability can act on both short and long time scales relative to the fundamental frequency of the given structure. For example, changing temperatures of wind turbine towers affecting their natural frequencies works on the order of hours and days (Hu et al., 2015), which in this context are considered



long-term effects. By contrast, complex aero–servo–elastic interactions of multi-megawatt wind turbines can cause short-term

variability of especially damping over a few minutes, due to dependencies on, e.g., rotor speed, blade pitch angle, and wind

speed (Hansen et al., 2006b, a).

Estimating aeroelastic (or operational) damping accurately is important for the improving design of multi-megawatt wind

turbines, as it is key design parameter for modelling fatigue and aeroelastic instabilities, i.e., Stall-Induced Vibrations (SIV) and

Vortex-Induced Vibrations (VIV) (Veers et al., 2023). Improved aeroelastic damping estimation may therefore enable designs

associated with less risk or less material usage. However, "a precise evaluation of aeroelastic damping remains an elusive

goal in some operating conditions" as stated in *Grand Challenges in the Design, Manufacture, and Operation of Future Wind

Turbine Systems* (Veers et al., 2023, p. 1090). The combination of nonstationary EOCs and EOC-sensitive damping complicates

the task of obtaining precise aeroelastic damping estimates for wind turbines.

OMA covers a broad class of output-only system identification methods used for estimating modal parameters for structures

in operating conditions, where the input (i.e., forcing or excitation) is not measured. Standard OMA techniques, for instance

the covariance-driven Stochastic Subspace Identification (COV-SSI) (Peeters and Roeck, 2001; Brincker and Ventura, 2015),

typically assumes the system is Linear and Time Invariant (LTI) with input resembling stationary white noise, and requires long-

time measurements. Brincker and Ventura (2015) suggest a minimum measurement time of $\frac{10}{f\zeta}$, where $f\zeta$ is the lowest natural

frequency–damping ratio multiple of a given structure. This translates to a measurement time requirement of approximately 28

minutes for a mode with natural frequency of 0.2 Hz and 3 % damping ratio, which is in the range of a multi-megawatt wind

turbine mode. The capabilities in tracking short-term variations in damping for these methods are therefore limited.

When identifying modal damping from output-only measurements, the *effect* of input on the measured output must be

accounted for, since both input *and* damping govern the near-resonant vibration amplitudes. This can be done by either elim-

inating the effect (i.e., averaging it out) or by modelling it (Au, 2017). Standard OMA methods rely on the former approach.

For instance, in COV-SSI covariance Toeplitz matrices of the output measurements are computed and used as equivalent free

response approximates of an assumed LTI system (Peeters and Roeck, 2001). However, a considerable amount of data is re-

quired (minimum measurement time of $\frac{10}{f\zeta}$) for the covariance Toeplitz matrices to be adequately estimated. Consequently, this

approach involves a trade-off between temporal resolution and estimate accuracy, which in the context of short-term variability

can be a limiting factor.

Nonstationary Auto-Regressive Moving Average (ARMA) time series models offer an avenue for accounting for nonstation-

ary input and time-varying system characteristics including modal parameters. ARMA models closely resembles the mathemat-

ical structure of discrete-time equations of motion, where the Auto-Regressive (AR) part plays the role of the left-hand (homo-

geneous) side and the AR *model* coefficients carry information of the *modal* parameters. Similarly, the Moving Average (MA)

part resembles the input, thus filtering out the effect of the excitation from the modal parameters. In Time-dependent Auto-

Regressive Moving Average (TARMA) models, the model coefficients and consequently the corresponding modal parameters,

are allowed to vary in time. One type is the Smoothness Priors TARMA (SP-TARMA) model, where the model coefficients

are modelled as autocorrelated stochastic (random) variables, for which evolution in time is constrained by smoothness priors

(Poulimenos and Fassois, 2006; Spiridonakos et al., 2010; Kitagawa and Gersch, 1996). The model coefficients of SP-TARMA





model are estimated locally in time, e.g., with a Kalman filter. The SP-TARMA models may be capable of tracking general
nonstationary signals but has limited capabilities in tracking abrupt or short-term changes (Poulimenos and Fassois, 2006).
Functional Series TARMA (FS-TARMA) models constitute TARMA type models whose model coefficients are represented
by predefined basis functions, allowing AR- and MA-coefficients to evolve deterministically in time. FS-TARMA models are
fitted to data by estimating *global* projection coefficients for each basis function, which minimize the model prediction errors.
If the basis functions can capture the time-varying nature of the system, FS-TARMA models are able to track abrupt and short-
term changes of the system. However, modal parameter trajectories in time for complex systems under EOC variability might
not lend themselves to be represented by a reasonable number of time-dependent basis functions, resulting in an intractable
number of parameters to be estimated.

One approach to capture the effects of changing EOCs on vibrating structures is to embed measured Environmental and
Operational Variables (EOVs) into the model. Various approaches have been proposed for this. Multi-megawatt wind turbines
pose a particular challenge, due to the intricate aero–servo–elastic interactions. Bogoevska et al. (2017) expands SP-TARMA
model residuals with a Polynomial Chaos Expansion (PCE) to account for long-term EOC variability, to improve accuracy of
structural health monitoring (SHM) of wind turbines. Avendaño-Valencia et al. (2017) introduces a Linear Parameter Varying
AR (LPV-AR) model to capture the short-term dynamics, and Gaussian Process (GP) regression is used to account for long-
term variability associated with changing wind speed in terms of 10-minute averages, which is extended from single-output
(univariate) to a multiple-output (multivariate) model by Avendaño-Valencia et al. (2020); Avendaño-Valencia and Chatzi
(2020), which, e.g., enables mode shape identification.

The present work concerns short-term damping (and natural frequency) estimation based on output-only measurements for
structures influenced by short-term varying EOCs, where short-term is in the order of seconds. The aforementioned methods are
based on models which are conditioned on (e.g., 10 minute) *statistics*, which limits their ability in tracking short-term changes.
The nonstationary and EOV-dependent GP-TARMA model (introduced in Ebbehøj et al. (2023)) is therefore conditioned on
EOV *time series*. The GP-TARMA model combines a FS-TARMA model where the basis functions may depend on multiple
EOV time series with Gaussian Processes, by modelling the projection coefficients for the basis functions as Gaussian rather
than deterministic variables, to allow for better representation of unaccounted disturbances. The capabilities of the GP-TARMA
model to track EOC variability is limited by the extent of how well the basis functions capture the nonstationary of the response
(e.g., slow or fast variations), and fundamentally by the measurement sampling rate.

Verification and validation are integral parts in establishing any new method or model in structural dynamics, and it is
especially important for output-only damping estimation methods, due to the latent and elusive nature of damping. This work
contributes in particular experimental validation of the GP-TARMA approach for short-term damping estimation, suitable for
application to wind turbines. The method is validated using vibration measurements from two distinctly different experimental
setups: a laboratory shear frame with abruptly changing damping realized with electromagnetic dampers, and a full-scale
7 MW wind turbine prototype deliberately driven to flutter-like instabilities (measurements published by Volk et al. (2020)).

The paper is structured as follows: Section 2 summarizes the details necessary for using the GP-TARMA model for short-
term modal parameter estimation from Ebbehøj et al. (2023), including the GP-TARMA model definition and estimation, a





model structure identification scheme, a model validation procedure, and downstream analysis of an estimated GP-TARMA

model. Section 3 presents the laboratory shear frame test setup, experimental procedures, and related analysis, results, and

discussion. In sect. 4, analysis of the full-scale multi-megawatt wind turbine instability measurements is performed, and the

results are presented and discussed.

## 2    Methods

This section provides a summary of the procedures for estimating short-term damping ratios (and natural frequencies) from

output-only measurements using a GP-TARMA model, which is introduced and presented in detail in Ebbehøj et al. (2023).

Necessary details for using the method are given, including the GP-TARMA model definition, procedures for estimation of

model parameters, selecting an appropriate model structure, and extracting modal parameters, and their uncertainties. The

entire procedure is summarized in Table 2.

### 2.1    The GP-TARMA model

The GP-TARMA model introduced in Ebbehøj et al. (2023) can be used to model a nonstationary, discrete-time (displace-

ment/velocity/acceleration) response $y_t \in \mathbb{R}$ influenced by EOCs, which can be described using $m$ EOVs $\boldsymbol{\xi}_t \in \mathbb{R}^{1 \times m}$, where

subscript $t$ denotes the time index, and $y_t$ and $\boldsymbol{\xi}_t$ are defined for $t = 1, \ldots, N$. The GP-TARMA model is closely related to

an FS-TARMA model (Poulimenos and Fassois, 2006; Spiridonakos and Fassois, 2014), for which the ARMA coefficients

evolves in time as trajectories spanned by EOV-dependent basis functions. The GP-TARMA model extends FS-TARMA by

modelling the basis function coefficients as Gaussian variables:

$$y_t + \underbrace{\sum_{i=1}^{n_a} a_i(\boldsymbol{\xi}_t) y_{t-i}}_{\text{AR-part}} = e_t + \underbrace{\sum_{i=1}^{n_c} c_i(\boldsymbol{\xi}_t) e_{t-i}}_{\text{MA-part}}, \quad e_t \sim \mathcal{N}\left(0, \sigma_{e,t}^2\right), \tag{1}$$

$$a_i(\boldsymbol{\xi}_t) = \sum_{j=1}^{p_a} u_{a_{i,j}} g_j(\boldsymbol{\xi}_t), \quad u_{a_{i,j}} \sim \mathcal{N}\left(\mu_{a_{i,j}}, \sigma_{a_{i,j}}^2\right), \tag{2}$$

$$c_i(\boldsymbol{\xi}_t) = \sum_{j=1}^{p_c} u_{c_{i,j}} h_j(\boldsymbol{\xi}_t), \quad u_{c_{i,j}} \sim \mathcal{N}\left(\mu_{c_{i,j}}, \sigma_{c_{i,j}}^2\right), \tag{3}$$

where $e_t$ is a Normally and Independently Distributed (NID) zero-mean innovations process with variance $\sigma_{e,t}^2$ which may

be time-varying, $a_i(\boldsymbol{\xi}_t)$ and $c_i(\boldsymbol{\xi}_t)$ are the $i^{\text{th}}$ EOV-dependent AR and MA coefficients, and $n_a$ ($n_c$) denotes the AR (MA)

model order. The AR and MA coefficients $a_i(\boldsymbol{\xi}_t)$ and $c_i(\boldsymbol{\xi}_t)$ are linear combinations of basis functions $g_j(\boldsymbol{\xi}_t)$ and $h_j(\boldsymbol{\xi}_t)$ and

the associated Gaussian projection coefficients $u_{a_{i,j}}$ and $u_{c_{i,j}}$. The complete set of AR (MA) basis functions for the full time

series $t = 1, \ldots, N$, constitute a functional subspace defined as:

$$\mathcal{F}_{AR} = \{\mathbf{g}_1(\boldsymbol{\xi}), \mathbf{g}_2(\boldsymbol{\xi}), \mathbf{g}_3(\boldsymbol{\xi}), \ldots, \mathbf{g}_{p_a}(\boldsymbol{\xi})\}, \tag{4a}$$

$$\mathcal{F}_{MA} = \{\mathbf{h}_1(\boldsymbol{\xi}), \mathbf{h}_2(\boldsymbol{\xi}), \mathbf{h}_3(\boldsymbol{\xi}), \ldots, \mathbf{h}_{p_c}(\boldsymbol{\xi})\}, \tag{4b}$$



where $\mathbf{g}_j(\boldsymbol{\xi}) = [g_j(\boldsymbol{\xi}_1), \dots, g_j(\boldsymbol{\xi}_N)]^T \in \mathbb{R}^{N \times 1}$ and $\mathbf{h}_j(\boldsymbol{\xi}) = [h_j(\boldsymbol{\xi}_1), \dots, h_j(\boldsymbol{\xi}_N)]^T \in \mathbb{R}^{N \times 1}$ are the $j^{\text{th}}$ basis functions for time indices $t = 1, \dots, N$, and $\boldsymbol{\xi} \in \mathbb{R}^{N \times m}$ contains the EOVs at the corresponding time indices. The basis functions $\mathbf{g}_j$ and $\mathbf{h}_j$ may be members of different orthogonal function families (e.g., Legendre polynomials or trigonometric functions), and depend on different EOVs (e.g., rotor speed, wind speed, and pitch angle). A *basis function type* consists of a function family and an EOV. The functional subspace $\mathcal{F}_{AR}$ ($\mathcal{F}_{MA}$) may consist of $k_a$ ($k_c$) basis function types, each with a basis order $p_{a_k}$ ($p_{c_k}$).

For example, $\mathcal{F}_{AR}$ could consist of a constant-valued bias vector $\mathbf{g}_1$, first and second order Legendre polynomials in wind speed $\mathbf{v}$, $\mathbf{g}_2(\mathbf{v})$ and $\mathbf{g}_3(\mathbf{v})$, and a first order Legendre polynomial in rotor speed $\boldsymbol{\Omega}$, $\mathbf{g}_4(\boldsymbol{\Omega})$. This would correspond to $k_a = 2$ basis function types (neglecting the trivial constant-valued bias type) with basis orders of $p_{a_1} = 2$ and $p_{a_2} = 1$ for the Legendre polynomials in wind speed and rotor speed, respectively. The orthogonality among basis functions in the functional subspaces should be ensured, using, e.g., modified Gram-Schmidt orthogonalization (Stewart, 2013). In cases where EOVs contain high-frequency (i.e., much higher than the frequency of highest mode) scatter, it can be beneficial to zero-phase low-pass filter (e.g., using MATLAB's `filtfilt` function) the EOVs to prevent the high-frequency scatter from propagating to the modal parameters (Ebbehøj et al., 2023).

ARMA models are closely linked with discrete-time Equations Of Motion (EOM). The AR-part resemble the left-hand side of discrete-time EOMs, which means the AR coefficients carries the physical characteristics of the system it models, i.e., natural frequencies and damping ratios in this case. The MA-part resembles the right-hand side of discrete-time EOMs, in the sense that it can capture the effect of stochastic excitation on the measured response $y_t$. Consequently, $\mathcal{F}_{AR}$ should be composed such that it can represent the time-varying nature of the natural frequencies and damping ratios, and $\mathcal{F}_{MA}$ should be selected to account for nonstationary stochastic excitation; examples of this are given in sects. 3.2 and 4.1.





Equations (1)–(3) can be expressed in the more compact regression form:

$$
y_t = \underbrace{\begin{bmatrix} -g_1(\boldsymbol{\xi}_t)\,y_{t-1} \\ -g_2(\boldsymbol{\xi}_t)\,y_{t-1} \\ \vdots \\ -g_{p_a}(\boldsymbol{\xi}_t)\,y_{t-1} \\ -g_1(\boldsymbol{\xi}_t)\,y_{t-2} \\ -g_2(\boldsymbol{\xi}_t)\,y_{t-2} \\ \vdots \\ -g_{p_a}(\boldsymbol{\xi}_t)\,y_{t-n_a} \\ h_1(\boldsymbol{\xi}_t)\,e_{t-1} \\ h_2(\boldsymbol{\xi}_t)\,e_{t-1} \\ \vdots \\ h_{p_c}(\boldsymbol{\xi}_t)\,e_{t-1} \\ h_1(\boldsymbol{\xi}_t)\,e_{t-2} \\ h_2(\boldsymbol{\xi}_t)\,e_{t-2} \\ \vdots \\ h_{p_c}(\boldsymbol{\xi}_t)\,e_{t-n_c} \end{bmatrix}^T}_{\boldsymbol{\Phi}_t^T} \underbrace{\begin{bmatrix} u_{a_{1,1}} \\ u_{a_{1,2}} \\ \vdots \\ u_{a_{1,p_a}} \\ u_{a_{2,1}} \\ u_{a_{2,2}} \\ \vdots \\ u_{a_{n_a,p_a}} \\ u_{c_{1,1}} \\ u_{c_{1,2}} \\ \vdots \\ u_{c_{1,p_c}} \\ u_{c_{2,1}} \\ u_{c_{2,2}} \\ \vdots \\ u_{c_{n_c,p_c}} \end{bmatrix}}_{\boldsymbol{\theta}} + e_t = \boldsymbol{\Phi}_t^T\,\boldsymbol{\theta} + e_t, \quad e_t \sim \mathcal{N}\left(0, \sigma_{e,t}^2\right),
\tag{5}
$$

where the *regression vector* $\boldsymbol{\Phi}_t = \boldsymbol{\Phi}_t(\boldsymbol{\xi}_t) \in \mathbb{R}^{(n_a p_a + n_c p_c) \times 1}$ contains regressed response measurements $y_{t-i}$ and innovations $e_{t-i}$ multiplied with appropriate basis function values, and the Gaussian projection coefficients are collected in $\boldsymbol{\theta} \in \mathbb{R}^{(n_a p_a + n_c p_c) \times 1}$. The GP-TARMA model for the full time series can be written in regression form, by employing stacked response and innova-

tions vectors $\mathbf{y} = [y_{1+n_m}, \ldots, y_N]^T \in \mathbb{R}^{(N-n_m) \times 1}$ and $\mathbf{e} = [e_{1+n_m}, \ldots, e_N]^T \in \mathbb{R}^{(N-n_m) \times 1}$:

$$
\mathbf{y} = \boldsymbol{\Phi}^T \boldsymbol{\theta} + \mathbf{e},
\tag{6}
$$

where $n_m = \max([n_a, n_c])$ denotes the maximum model order, and $\boldsymbol{\Phi} \in \mathbb{R}^{(n_a p_a + n_c p_c) \times (N-n_m)}$ is the *regression matrix*.

For a given dataset $\mathcal{D} = \{\mathbf{y}, \boldsymbol{\xi}\}$, the GP-TARMA model is fully characterised by $\mathcal{M} = \{\mathcal{S}, \mathcal{P}\}$, where $\mathcal{S} = \{\mathcal{F}_{AR}, \mathcal{F}_{MA}, n_a, n_c\}$ contains the *model structure* and $\mathcal{P} = \{\boldsymbol{\mu}_{\boldsymbol{\theta}}, \boldsymbol{\Sigma}_{\boldsymbol{\theta}}, \boldsymbol{\Sigma}_{\mathbf{e}}\}$ contains the time-varying innovations variance $\boldsymbol{\Sigma}_{\mathbf{e}} =$

$\mathrm{diag}([\sigma_{e,1+n_m}^2, \ldots, \sigma_{e,N}^2]) \in \mathbb{R}^{(N-n_m) \times (N-n_m)}$, and the hyper-parameters for the Gaussian projection coefficients, i.e., the means $\boldsymbol{\mu}_{\boldsymbol{\theta}} \in \mathbb{R}^{(n_a p_a + n_c p_c) \times 1}$ and covariance $\boldsymbol{\Sigma}_{\boldsymbol{\theta}} \in \mathbb{R}^{(n_a p_a + n_c p_c) \times (n_a p_a + n_c p_c)}$.

## 2.2   Model parameter estimation

A procedure for Maximum Likelihood (ML) estimation of the hyper-parameters and innovations variance in $\mathcal{P}$ (via the Expectation Maximization algorithm) presented in Ebbehøj et al. (2023) (see also Avendaño-Valencia et al. (2017)) is summarized

in this section.





With the model parameters $\boldsymbol{\theta}$ and response $\mathbf{y}$ in Eq. (6) modelled as random variables, the Probability Density Function (PDF) of their joint distribution governs the probability of the two (model parameters and response) occurring simultaneously:

$$p(\mathbf{y}, \boldsymbol{\theta} | \boldsymbol{\Phi}, \mathcal{P}) = p(\boldsymbol{\theta} | \mathbf{y}, \boldsymbol{\Phi}, \mathcal{P}) \, p(\mathbf{y} | \boldsymbol{\Phi}, \mathcal{P}), \tag{7}$$

where $p(\boldsymbol{\theta} | \mathbf{y}, \boldsymbol{\Phi}, \mathcal{P})$ denotes the *posterior distribution* of the model parameters, and $p(\mathbf{y} | \boldsymbol{\Phi}, \mathcal{P})$ the *marginal likelihood of the response*. For the GP-TARMA model in Eq. (6), these distributions are Gaussian, and can be specified as (Ebbehøj et al., 2023) (see also Murphy (2023), Ch. 2, and Avendaño-Valencia et al. (2017)):

$$p(\boldsymbol{\theta} | \mathbf{y}, \boldsymbol{\Phi}, \mathcal{P}) = \mathcal{N}\left(\hat{\boldsymbol{\theta}}, \hat{\mathbf{P}}_{\boldsymbol{\theta}}\right), \tag{8}$$

$$p(\mathbf{y} | \boldsymbol{\Phi}, \mathcal{P}) = \mathcal{N}\left(\boldsymbol{\Phi}^T \boldsymbol{\mu}_{\boldsymbol{\theta}}, \boldsymbol{\Sigma}_{\boldsymbol{\varepsilon}}\right), \tag{9}$$

where

$$\hat{\boldsymbol{\theta}} = \mathbb{E}\left[\boldsymbol{\theta} | \mathbf{y}, \boldsymbol{\Phi}, \mathcal{P}\right] = \boldsymbol{\mu}_{\boldsymbol{\theta}} + \mathbf{K}\left(\mathbf{y} - \boldsymbol{\Phi}^T \boldsymbol{\mu}_{\boldsymbol{\theta}}\right), \tag{10a}$$

$$\hat{\mathbf{P}}_{\boldsymbol{\theta}} = \mathbb{E}\left[(\boldsymbol{\theta} - \hat{\boldsymbol{\theta}})(\boldsymbol{\theta} - \hat{\boldsymbol{\theta}})^T | \mathbf{y}, \boldsymbol{\Phi}, \mathcal{P}\right] = \boldsymbol{\Sigma}_{\boldsymbol{\theta}} - \mathbf{K}\,\boldsymbol{\Phi}^T\boldsymbol{\Sigma}_{\boldsymbol{\theta}}, \tag{10b}$$

$$\mathbf{K} = \boldsymbol{\Sigma}_{\boldsymbol{\theta}}\,\boldsymbol{\Phi}\,\boldsymbol{\Sigma}_{\boldsymbol{\varepsilon}}^{-1}, \tag{10c}$$

$$\boldsymbol{\Sigma}_{\boldsymbol{\varepsilon}} = \boldsymbol{\Sigma}_{\mathbf{e}} + \boldsymbol{\Phi}^T\boldsymbol{\Sigma}_{\boldsymbol{\theta}}\boldsymbol{\Phi}, \tag{10d}$$

where $\mathbb{E}[X]$ is the expected value of $X$, $\boldsymbol{\Sigma}_{\boldsymbol{\varepsilon}} \in \mathbb{R}^{(N-n_m)\times(N-n_m)}$ is the covariance matrix of the *prior* (i.e., before observing the actual response) one-step-ahead prediction errors, and the innovations variance for each time index is collected in $\boldsymbol{\Sigma}_{\mathbf{e}} = \mathrm{diag}\left(\sigma_{1+nm}^2, \ldots, \sigma_N^2\right) \in \mathbb{R}^{(N-n_m)\times(N-n_m)}$. The mean and covariance of the posterior parameter distribution, $\hat{\boldsymbol{\theta}} \in \mathbb{R}^{(n_a p_a + n_c p_c)\times 1}$ and $\hat{\mathbf{P}}_{\boldsymbol{\theta}} \in \mathbb{R}^{(n_a p_a + n_c p_c)\times(n_a p_a + n_c p_c)}$, are also referred to as the *Maximum A Posteriori* (MAP) estimates of $\boldsymbol{\mu}_{\boldsymbol{\theta}}$ and $\boldsymbol{\Sigma}_{\boldsymbol{\theta}}$, respectively.

The marginal likelihood of the response Eq. (9) also serves as the marginal likelihood of the hyper-parameters $\mathcal{L}(\mathcal{P}|\mathbf{y}, \boldsymbol{\Phi})$, meaning that the hyper-parameters in $\mathcal{P}$ can be estimated, by maximizing the marginal hyper-parameter likelihood. This forms a ML optimization problem, which can be formulated in terms of the log-likelihood as (Avendaño-Valencia et al., 2017; Murphy, 2023; Rasmussen and Williams, 2006):

$$\hat{\mathcal{P}} = \underset{\mathcal{P}}{\arg\max}\, \ln\mathcal{L}\left(\mathcal{P}|\mathbf{y}, \boldsymbol{\Phi}\right), \tag{11a}$$

$$\text{where} \quad \ln\mathcal{L}\left(\mathcal{P}|\mathbf{y}, \boldsymbol{\Phi}\right) = -\frac{N}{2}\ln 2\pi - \frac{1}{2}\left(\ln|\boldsymbol{\Sigma}_{\boldsymbol{\varepsilon}}| + \boldsymbol{\varepsilon}^T\boldsymbol{\Sigma}_{\boldsymbol{\varepsilon}}^{-1}\boldsymbol{\varepsilon}\right), \tag{11b}$$

where the vector $\boldsymbol{\varepsilon} = [\varepsilon_{1+nm}, \ldots, \varepsilon_N]^T \in \mathbb{R}^{(N-n_m)\times 1}$ contains the prior one-step-ahead prediction errors computed at time index $t$ as:

$$\varepsilon_t = y_t - \boldsymbol{\Phi}_t^T \boldsymbol{\mu}_{\boldsymbol{\theta}}. \tag{12}$$

Solutions to the ML optimization can be approximated using the general-purpose Expectation Maximization (EM) algorithm, which constitutes an expectation step (E-step) and a Maximization step (M-step), which are iterated until convergence. During





the E-step, the posterior (expected) mean and covariance of the model parameters, $\hat{\boldsymbol{\theta}}$ and $\hat{\mathbf{P}}_{\boldsymbol{\theta}}$, are evaluated using Eqs. (10a)–(10d) based on the hyper-parameters from previous EM-iteration. In the M-step, the hyper-parameters are updated using the model parameter posterior mean and covariance from the E-step by the explicit update equations (Avendaño-Valencia et al., 2017):

$$\boldsymbol{\mu_\theta}^{(i)} = \hat{\boldsymbol{\theta}}^{(i-1)}, \tag{13a}$$

$$\boldsymbol{\Sigma_\theta}^{(i)} = \boldsymbol{\delta}\,\boldsymbol{\delta}^T + \hat{\mathbf{P}}_{\boldsymbol{\theta}}^{(i-1)}, \tag{13b}$$

$$\boldsymbol{\delta} = \hat{\boldsymbol{\theta}}^{(i-1)} - \boldsymbol{\theta}^{(i-1)}, \tag{13c}$$

$$\hat{\boldsymbol{\Sigma}}_{\mathbf{e}}^{(i)} = \mathrm{diag}([\hat{\sigma}_{e,1}^2, \ldots, \hat{\sigma}_{e,N}^2]), \tag{13d}$$

where $^{(i)}$ denotes the designated variable is from the $i^{\text{th}}$ EM-iteration. The time-varying innovations variance $\hat{\boldsymbol{\Sigma}}_{\mathbf{e}} \in \mathbb{R}^{(N-n_m)\times(N-n_m)}$ can be estimated using a $2K+1$ sample moving window:

$$\hat{\sigma}_{e,t}^2 = \frac{1}{2K}\sum_{\tau=t-K}^{t+K}(e_\tau - \hat{\mu}_{e,t})^2, \quad \hat{\mu}_{e,t} = \frac{1}{2K+1}\sum_{\tau=t-K}^{t+K}e_\tau, \tag{14}$$

where $e_\tau = y_\tau - \left(\boldsymbol{\Phi}^{(i-1)}\right)^T \hat{\boldsymbol{\theta}}^{(i-1)}$, and $\hat{\mu}_{e,t}$ is the innovations mean for the window with time index $t$. Although the latter is assumed to be zero, it is included in Eq. (14), because it has been observed to improve accuracy of damping ratio estimates in cases where the response measurements are influenced by stochastic excitation with time-varying mean value. The time-invariant innovations variance can likewise be estimated using Eq. (14) with $K = N/2$.

The EM-algorithm has been shown to converge to a *local* likelihood maximum, not necessarily *global* (e.g., Bishop (2006)). Consequently, accurate initial values of the hyper-parameters $\mathcal{P}^{(0)}$ are required. This can be obtained by estimating the model parameters for the corresponding FS-TARMA model, where $u_{a_{i,j}}$ and $u_{c_{i,j}}$ are *scalars* rather than Gaussian variables, and using these parameter estimates as initial hyper-parameter values $\mathcal{P}^{(0)}$. The corresponding FS-TARMA model parameters can be estimated using, e.g., the two-stage least squares (2SLS) or the two-stage weighted least squares (2SWLS) method (Spiridonakos and Fassois, 2014; Poulimenos and Fassois, 2006). Convergence of the EM-algorithm is implied by consistent and small changes in $\ln \mathcal{L}(\mathcal{P}|\mathbf{y},\boldsymbol{\Phi})$, and in the hyper-parameters $\mathcal{P}$ over EM-iterations.

### 2.3 Model validation

Once a GP-TARMA model is estimated, it is important to validate that model adequately represents the observations (Poulimenos and Fassois, 2006; Spiridonakos and Fassois, 2014; Madsen, 2007). Model validation in this present work (as in Ebbehøj et al. (2023)) consists of *residual analysis* and *cross validation*. Residual analysis tests whether the model residuals $e_t$ (i.e., the innovations) satisfies the NID assumption, i.e., whether they resemble white noise. A standard whiteness test is to check whether the Auto-Correlation Function (ACF) of the residuals resembles that of white noise, i.e., significantly uncorrelated at time lags $\tau > 0$. However, this is not generally applicable to the case of nonstationary residuals, as these would be correlated through the time-varying innovations variance. An approach to partially circumvent this issue is to normalize the residuals with



the estimated time-varying innovations variance $\hat{\sigma}_{e,t}^2$ (Fouskitakis and Fassois, 2002):

$$z_t = \frac{e_t}{\hat{\sigma}_{e,t}^2}, \quad t = 1 + n_m, \ldots, N. \tag{15}$$

For nonstationary residuals, the ACF test is sensitive to the accuracy of the time-varying innovations variance estimate, and should thus be supplemented by a test insensitive to the innovations variance estimate.

An alternative whiteness test, fully applicable to the nonstationary case, is a simple *sign test*. Sign changes of a zero mean white noise sequence are expected to occur (on average) every other time step, and governed by the binomial distribution, which is approximated by the Gaussian distribution for large $N_{seq}$ (Madsen, 2007):

$$\text{Number of changes in sign} \in \mathcal{B}\left(N_{seq} - 1, \frac{1}{2}\right) \approx \mathcal{N}\left(\mu_{wn}, \sigma_{wn}^2\right), \tag{16}$$

where $N_{seq}$ is the number of samples in the residual sequence, and the mean and variance are given by $\mu_{wn} = (N_{seq} - 1)/2$ and $\sigma_{wn}^2 = (N_{seq} - 1)/4$, respectively. Whether a residual sequence $\{e_t : t = 1 + n_m, \ldots, N\}$ resembles white noise can thus be tested by checking whether the number of sign changes adheres to Eq. (16) under some significance level.

Cross validation is performed by splitting a dataset of a single recording in a training and a test set containing $75\%$ and $25\%$ of the data points, respectively. The model is solely estimated using the training set, and is subsequently tested in terms of residual tests and whether the orders of magnitude of prediction errors are the same for the training and test set. If the prediction errors from the training set are much smaller than from the test set, it suggests the model is over-fitted, i.e., the model excessively represents the measured realization of the stochastic response, rather than the underlying system. However, for the nonstationary case, it is only applicable if the two sets have comparable characteristics.

## 2.4 Model structure identification

In this section a procedure for identifying a suitable model structure, i.e., $\mathcal{S} = \{\mathcal{F}_{AR}, \mathcal{F}_{MA}, n_a, n_c\}$, is summarized (see Ebbehøj et al. (2023) for details). A *suitable model structure* is sufficiently complex to capture the underlying system producing the response measurements, while avoiding over-fitting. For identifying a suitable model structure, a range of candidate models with different model structures are estimated, and compared in terms of predictive performance and capability of capturing the vibration modes of interest.

To compare the predictive performance (i.e., the prior one-step-ahead prediction errors) of the candidate models, the residual sum of squares normalized by the series sum of squares (RSS/SSS) and the Bayesian Information Criteria (BIC) are used. The RSS/SSS is given by:

$$RSS/SSS = \sum_{t=1+n_m}^{N} \varepsilon_t^2 \Big/ \sum_{t=1+n_m}^{N} y_t^2, \tag{17}$$

where $\varepsilon$ is the prior prediction error defined in Eq. (12). The BIC is given by:

$$BIC = -\ln \mathcal{L}(\mathcal{P}|\mathbf{y}, \mathbf{\Phi}) + \frac{\ln N}{2} d \tag{18}$$



**Table 1.** Model structure identification procedure (adapted from Ebbehøj et al. (2023)); high order or dimensionality denoted by $^*$

---

1) Propose candidate functional subspaces $\mathcal{F}_{AR}^*$ and $\mathcal{F}_{MA}^*$

2) For $n_c^*$, $\mathcal{F}_{AR}^*$, and $\mathcal{F}_{MA}^*$: Find optimal AR model order $\hat{n}_a \in \{1, 2, \dots, n_a^*\}$

3) For $\hat{n}_a$, $\mathcal{F}_{AR}^*$, and $\mathcal{F}_{MA}^*$: Find optimal MA model order $\hat{n}_c \in \{1, 2, \dots, n_c^*\}$

4) For $\hat{n}_a$, $\hat{n}_a$, and $\mathcal{F}_{MA}^*$: Find optimal basis order $\hat{p}_{a_k}$ for the $k^{\text{th}}$ AR basis function type. Repeat for all AR basis function types $k = 1, \dots, k_a$ to obtain $\hat{\mathcal{F}}_{AR}$

5) For $\hat{n}_a$, $\hat{n}_c$, and $\hat{\mathcal{F}}_{AR}$: Find optimal basis order $\hat{p}_{c_k}$ for the $k^{\text{th}}$ MA basis function type. Repeat for all MA basis function types $k = 1, \dots, k_c$ to obtain $\hat{\mathcal{F}}_{MA}$

---

Steps 1) – 5) may be repeated for other combinations of basis function types, i.e., testing other EOVs and/or family function types

---

where $d$ is the number of independently adjusted parameters. RSS/SSS and BIC both measure the prediction errors, but BIC also penalises model complexity, i.e., the number of model parameters. The required model complexity for capturing the vibration modes of interest are identified using frequency stabilization diagrams, as commonly used for time domain OMA methods (Brincker and Ventura, 2015; Peeters and Roeck, 2001; Avendaño-Valencia et al., 2017). Typically, the predictive performance converges at lower model orders compared to the model orders required to capture the modes of interest, i.e., the latter is typically the driving selection criteria.

A simple backward regression scheme is employed to identify a suitable model structure, i.e., starting with high model orders $n_a^*$ and $n_c^*$, and functional subspaces of high dimensionality $\mathcal{F}_{AR}^*$ and $\mathcal{F}_{MA}^*$, ensuring sufficient model complexity to capture the nonstationary response. The complexity of the initial model is then reduced in four stages: First models with lower AR orders are estimated, and the most suitable AR order is identified. Then the optimal MA order (given the optimal AR order) is identified in similar fashion. The optimal basis orders for each AR and MA basis function type are identified in the same way. These stages are repeated for different combinations of AR and MA basis function types. The procedure is summarized in Table 1.

### 2.5 Estimating modal parameters and their uncertainties

In this section the necessary results for estimating "frozen" modal parameters (excluding mode shapes) from an estimated GP-TARMA model, and approximating the corresponding uncertainties are summarized. The frozen properties of a time-varying system represent the LTI properties at "frozen-time" $t'$, i.e., the time-varying system is represented by an infinite sequence of LTI systems (Poulimenos and Fassois, 2006). See Ebbehøj et al. (2023) for more details, and also Poulimenos and Fassois (2006); Avendaño-Valencia and Fassois (2014); Spiridonakos et al. (2010); Avendaño-Valencia et al. (2017) and Yang and Lam (2019) on computation of frozen modal parameters and their uncertainty, respectively.

The frozen modal parameters can be computed from the time-varying AR coefficients, since an equivalent discrete-time state-space model with system matrix $\mathbf{L}_{t'}$ can be formulated from the AR coefficients $\{a_i(\boldsymbol{\xi}_{t'}) : i = 1, \dots, n_a\}$ at each time-




step $t'$. First step is to compute the discrete-time eigenvalues of the eigenvalue problem:

$$\mathbf{L}_{t'}\mathbf{v}_i = \rho_i\mathbf{v}_i, \quad i = 1,2,\ldots,n_a, \tag{19}$$

where $t'$ is the "frozen-time" time index, and $\mathbf{v}_i = \mathbf{v}_{i,t'}$, and $\rho_i = \rho_{i,t'}$ are the $i^{\text{th}}$ right eigenvector and eigenvalue of $\mathbf{L}_{t'}$:

$$\mathbf{L}_{t'} = \begin{bmatrix} \begin{matrix} 0 \\ \vdots \\ \vdots \\ 0 \end{matrix} & \mathbf{I}_{(n_a-1)\times(n_a-1)} \\ \hline -a_{n_a}(\boldsymbol{\xi}_{t'}) & \cdots \quad -a_2(\boldsymbol{\xi}_{t'}) \quad -a_1(\boldsymbol{\xi}_{t'}) \end{bmatrix}_{n_a\times n_a}, \tag{20}$$

270 where $\mathbf{I}$ is the identity matrix with dimensions as indicated by the subscript.

The $i^{\text{th}}$ natural frequency $f_{i,t'}$ (in Hz) and damping ratio $\zeta_{i,t'}$ for frozen time index $t'$ can be computed by (Yang and Lam, 2019; Poulimenos and Fassois, 2006; Spiridonakos et al., 2010; Avendaño-Valencia et al., 2017):

$$f_{i,t'} = \frac{|\eta_{i,t'}|}{2\pi} \quad \text{and} \quad \zeta_{i,t'} = -\frac{\text{Re}(\eta_{i,t'})}{|\eta_{i,t'}|}, \tag{21a}$$

$$\text{where} \quad \eta_{i,t'} = f_s\ln(\rho_{i,t'}) \tag{21b}$$

275 is the equivalent $i^{\text{th}}$ continuous-time eigenvalue.

The uncertainty of the modal parameter estimates can be approximated by propagating the estimated AR coefficient uncertainty (quantified by $\hat{\mathbf{P}}_{\boldsymbol{\theta}}$) through the steps required to compute modal parameters from AR coefficients. These steps constitute obtaining discrete-time eigenvalues in Eq. (19), transforming the discrete-time eigenvalues to continuous-time eigenvalues in Eq. (21b) and computing the modal parameters from the continuous-time eigenvalues in Eq. (21a). The uncertainty propaga-

280 tion can be achieved through either Monte Carlo simulation, or analytically using the first-order delta method. The former is straightforward to implement but is computationally costly, whereas the latter offers a much smaller computational burden at the price of a more elaborate implementation.

The analytic uncertainty propagation method is only valid for *Gaussian* PDFs with *small variances*. The necessary results are stated below; see Ebbehøj et al. (2023) for a brief introduction and Yang and Lam (2019) for derivations and further details.

285 The uncertainty of the $i^{\text{th}}$ set of natural frequencies and damping ratios can be quantified by the variance:

$$\boldsymbol{\Sigma}_{f_i,\zeta_i} = \frac{\partial\mathbf{F}_{3,i}}{\partial\eta_i}\frac{\partial\mathbf{F}_{2,i}}{\partial\rho_i}\frac{\partial\mathbf{F}_{1,i}}{\partial\text{vec}(\mathbf{L}_{t'})}\text{vec}(\mathbf{P}_{\mathbf{L}_{t'}})\left(\frac{\partial\mathbf{F}_{1,i}}{\partial\text{vec}(\mathbf{L}_{t'})}\right)^T\left(\frac{\partial\mathbf{F}_{2,i}}{\partial\rho_i}\right)^T\left(\frac{\partial\mathbf{F}_{3,i}}{\partial\eta_i}\right)^T, \tag{22}$$

where $\boldsymbol{\Sigma}_{f_i,\zeta_i}\in\mathbb{R}^{2\times2}$, $\text{vec}(\cdot)$ is the vectorization operator transforming a $M\times N$ matrix $\mathbf{A} = [\mathbf{a}_1,\mathbf{a}_2,\ldots,\mathbf{a}_N]$ to a column vector $\text{vec}(\mathbf{A}) = [\mathbf{a}_1^T,\mathbf{a}_2^T,\ldots,\mathbf{a}_N^T]^T\in\mathbb{R}^{MN\times1}$, and the vectorized posterior variance matrix $\text{vec}(\mathbf{P}_{\mathbf{L}_{t'}})$ corresponding to $\mathbf{L}_{t'}$ is:

$$\text{vec}(\mathbf{P}_{\mathbf{L}_{t'}}) = \text{diag}(\text{vec}(\mathbf{P}_{\mathbf{L}_{t'}})), \quad \mathbf{P}_{\mathbf{L}_{t'}} = \begin{bmatrix} \mathbf{0}_{(n_a-1)\times(n_a)} \\ \hline \sigma_{a_{n_a}}^2(\boldsymbol{\xi}_{t'}) \quad \cdots \quad \sigma_{a_2}^2(\boldsymbol{\xi}_{t'}) \quad \sigma_{a_1}^2(\boldsymbol{\xi}_{t'}) \end{bmatrix}_{n_a\times n_a}, \tag{23}$$





where $\sigma^2_{a_i}(\boldsymbol{\xi}_{t'}) = \sum_{j=1}^{p_a} \sigma^2_{a_{i,j}} g_j(\boldsymbol{\xi}_{t'})^2$. Note that $\sigma^2_{a_i}(\boldsymbol{\xi}_{t'})$ is not necessarily Gaussian, as the individual terms of the sum can be dependent. The outputs of the three functions are:

$$\boldsymbol{\rho} = \mathbf{F}_1\left(\text{vec}(\mathbf{L}_{t'})\right), \tag{24a}$$

$$\boldsymbol{\eta} = \mathbf{F}_2(\boldsymbol{\rho}), \tag{24b}$$

$$\boldsymbol{\beta} = \mathbf{F}_3(\boldsymbol{\eta}), \tag{24c}$$

where $\boldsymbol{\rho} = [\rho_1, \rho_2, \ldots, \rho_{n_a}]^T$, $\boldsymbol{\eta} = [\eta_1, \eta_2, \ldots, \eta_{n_a}]^T$, and $\boldsymbol{\beta} = \{[f_{i,t'}, \zeta_{i,t'}]^T : i = 1, \ldots, n_a\}$. The partial derivative of $\mathbf{F}_{1,i}$ with respect to $\text{vec}(\mathbf{L}_{t'})$ is:

$$\frac{\partial \mathbf{F}_{1,i}}{\partial \text{vec}(\mathbf{L}_{t'})} = \begin{bmatrix} \text{Re}\left(\frac{\partial \rho_i}{\partial \text{vec}(\mathbf{L}_{t'})}\right) \\ \text{Im}\left(\frac{\partial \rho_i}{\partial \text{vec}(\mathbf{L}_{t'})}\right) \end{bmatrix}_{n_a^2 \times 1}, \quad \text{where} \quad \frac{\partial \rho_i}{\partial \text{vec}(\mathbf{L}_{t'})} = \frac{\mathbf{v}_i^T \otimes \mathbf{w}_i^\top}{\mathbf{w}_i^\top \mathbf{v}_i}, \tag{25}$$

where $\mathbf{a}^\top$ denotes the complex conjugate transpose of $\mathbf{a}$, $\otimes$ denotes Kronecker's product, and $\mathbf{w}_i$ is the $i^{\text{th}}$ left-eigenvectors of $\mathbf{L}_{t'}$ satisfying

$$\mathbf{w}_i^\top \mathbf{L}_{t'} = \rho_i \mathbf{w}_i^\top, \tag{26}$$

where $\mathbf{w}_i$ is a column vector. The partial derivatives of functions $\mathbf{F}_{2,i}$ and $\mathbf{F}_{3,i}$ are

$$\frac{\partial \mathbf{F}_{2,i}}{\partial \rho_i} = \frac{1}{T_s |\hat{\rho}_i|^2} \begin{bmatrix} \text{Re}(\hat{\rho}_i) & \text{Im}(\hat{\rho}_i) \\ -\text{Im}(\hat{\rho}_i) & \text{Re}(\hat{\rho}_i) \end{bmatrix}_{2\times 2} \quad \text{and} \quad \frac{\partial \mathbf{F}_{3,i}}{\partial \eta_i} = \frac{1}{|\hat{\eta}_i|} \begin{bmatrix} \dfrac{\text{Re}(\hat{\eta}_i)}{2\pi} & \dfrac{\text{Im}(\hat{\eta}_i)}{2\pi} \\ -\dfrac{\text{Im}(\hat{\eta}_i)^2}{|\hat{\eta}_i|^2} & \dfrac{\text{Re}(\hat{\eta}_i)\text{Im}(\hat{\eta}_i)}{|\hat{\eta}_i|^2} \end{bmatrix}_{2\times 2}, \tag{27}$$

where hats denote estimated values.

The procedure for computing natural frequencies and damping ratios, and their uncertainties from AR-coefficients is summarized in Table 2.

## 2.6 Considerations for practical implementation

In this section some considerations which may be useful for practical application of the GP-TARMA approach are listed. These are based on using the GP-TARMA model for various applications including wind turbine blade response of an operational wind turbine during an extreme coherent wind gust (BHawC simulation, see Ebbehøj et al. (2023)), field measurements from Siemens Gamesa's fleet, and the applications in the present paper:

- Reduce bandwidth of response (output): Low-pass filter and subsequently down-sample response prior to GP-TARMA modelling, such that the reduced bandwidth only contains the modes of interest, i.e., unimportant response components at higher frequencies are filtered out. *Effects:* Reduces computational cost of estimating model; may help reduce risk of over-fitting in cases with low amount of training data



**Table 2.** Procedure for short-term modal parameter estimation using GP-TARMA model (adapted from Ebbehøj et al. (2023))

**Required data**:

- Dynamic response $y_t$ and influencing EOVs $\boldsymbol{\xi}_t$ for $t = 1, \ldots, N$
- Define maximum number of EM iterations $N_{iter}$

**Model structure identification**:

- Use procedure summarized in Table 1

**Model estimation, initialization:**

- Compute set of initial hyper-parameters $\mathcal{P}^{(0)} = \left\{ \boldsymbol{\mu_\theta}^{(0)}, \boldsymbol{\Sigma_\theta}^{(0)}, \hat{\boldsymbol{\Sigma}}_{\mathbf{e}}^{(0)} \right\}$ using 2SLS estimates of corresponding FS-TARMA model

**Model estimation, EM iterations:**

- For $i = 1, \ldots, N_{iter}$

  - *Expectation step*: Compute posterior model parameter mean $\hat{\boldsymbol{\theta}}$ and covariance $\hat{\mathbf{P}}_{\boldsymbol{\theta}}$ based on hyper-parameters $\mathcal{P}^{(i-1)}$ using Eqs. (10a)–(10d)

  - *Maximization step*: Compute updated hyper-parameters $\mathcal{P}^{(i)} = \left\{ \boldsymbol{\mu_\theta}^{(i)}, \boldsymbol{\Sigma_\theta}^{(i)}, \hat{\boldsymbol{\Sigma}}_{\mathbf{e}}^{(i)} \right\}$ using Eqs. (13a)–(13d)

  - *Check convergence*: If either change in hyper-parameters or marginal log-likelihood over iteration is smaller than a specified tolerance, stop iterating, i.e., if $|\mathcal{P}^{(i)} - \mathcal{P}^{(i-1)}| < \epsilon_{\mathcal{P}}$ or $|\ln \mathcal{L}\left(\mathcal{P}^{(i)}|\mathbf{y}, \boldsymbol{\Phi}^{(i)}\right) - \ln \mathcal{L}\left(\mathcal{P}^{(i-1)}|\mathbf{y}, \boldsymbol{\Phi}^{(i-1)}\right)| < \epsilon_{\mathcal{L}}$

**Model validation**

- Use procedure outlined in sect. 2.3 related to Eqs. (15) and (16), proceed to downstream analysis only if residuals are adequately white

**Modal parameter estimation**:

- For $t' = 1 + n_a, \ldots N$:

  - Compute "frozen-time" natural frequencies $f_{i,t'}$ and damping ratios $\zeta_{i,t'}$ from AR-coefficients $a_i(\boldsymbol{\xi}_t)$ for $i = 1, \ldots, n_a$ using Eqs. (19), (20), (21a) and (21b)

  - Compute modal parameter uncertainty $\{\boldsymbol{\Sigma}_{f_i, \zeta_i} i = 1, \ldots, n_a\}$ analytically using Eqs. (22)–(27) or by Monte Carlo simulation

- Filter out high-frequency content of EOV signals by low-pass filtering. *Effects:* May allow for a better fit, as modal properties usually do not vary at high frequencies (many times per second); preventing high-frequency noise propagating from AR-coefficients to modal parameters

- Use zero-phase filtering for low-pass filtering of response and EOV signals. *Effect:* Response and EOV signals are not phase shifted by filtering.

- Add small amount of artificial zero-mean NID noise to (low-pass filtered) response signal prior to GP-TARMA modelling. Adequate RMS levels of noise relative to the signal are typically between 1 and 5 %. *Effect:* This is an example





of *jittering* and *data augmentation* in general. It can have a regularizing effect, i.e., reduce risk of over-fitting, and commonly used in the fields of, e.g., image analysis and for machine learning methods (Iwana and Uchida, 2021; Bishop, 1995; Ding et al., 2015; Erenler and Serinagaoglu Dogrusoz, 2019). According to Bishop (1995) adding noise to training
data is equivalent to Tikhonov regularization.

## 3   Laboratory test: Shear frame structure with time-varying damping

This section presents an experimental setup of a shear frame (SF) structure with controlled time-varying damping, and the experimental procedures used to generate response measurements for testing and validating the GP-TARMA approach. In Ebbehøj et al. (2023), the method is verified using responses from two simulated cases: a simple two-mode model response,
and an edgewise blade response during normal operation obtained with BHawC (Siemens Gamesa's in-house aeroelastic code). This section provides an experimental validation, based on measurements of a simple structure conducted in a controlled setting.

### 3.1   Experimental setup and procedures

An experimental test rig with a three-storey SF structure was used for testing modal parameter estimation methods for structures
with time-varying damping. The structure is equipped with a pair of voltage-controlled electromagnetic dampers (EMDs) at each floor. Two types of tests were conducted: a base-excitation test, where the SF structure was excited with a shaker table, and a hammer test, where the structure is excited with a (measured) hammer impact. Figure 1 illustrates the experimental setup and instrumentation, and Fig. 2 shows the SF structure. The floors are quadratic $(110 \times 110 \times 10)$ mm, constructed from aluminium with 2 mm thick copper plate inlays at the top and bottom to increase the effect of eddy current damping through EMDs ((A)
in Figs. 1 and 2). The beams separating the floors are $(150 \times 18 \times 0.5)$ mm, and are realized by four steel rulers (C) which are fixed to each floor.

As can be seen in Fig. 1b, shaker table (used for base-excitation tests) is driven by an amplified, digital signal using a National Instruments (NI) relay module (NI 9481) and LabView as interface with the computer. The EMDs are powered by a 24 V signal, which is controlled in an on/off manner using the NI relay module, which is controlled using FlexLogger software.
The displacement response at each floor is measured by laser displacement sensors, and sampled by a NI data acquisition (DAQ) module (NI 9215) via FlexLogger, along with relevant control signals, e.g., control signals for the EMDs and amplifier.

The input signal for the shaker table used for the base-excitation test, is a stochastic normally distributed broadband signal (i.e., low-pass filtered white noise), to achieve stochastic excitation of the first three modes of the SF structure. The test is run for 60 minutes, all signals are collected using a sampling frequency of 50 Hz, and all EMDs are turned on and off during the
test with periods of constant voltage ranging from 240 to 540 s.

For the base-excitation test, the displacement response at the third floor is used and referred to as $y(t)$ in sect. 3, and the measured voltage over the EMDs $v(t)$ is used as EOV to predict changes in damping caused by the EMDs. Figure 3 shows the displacement response $y(t)$, the corresponding spectrogram and spectrum, along with voltage over the EMDs $v(t)$, indicating



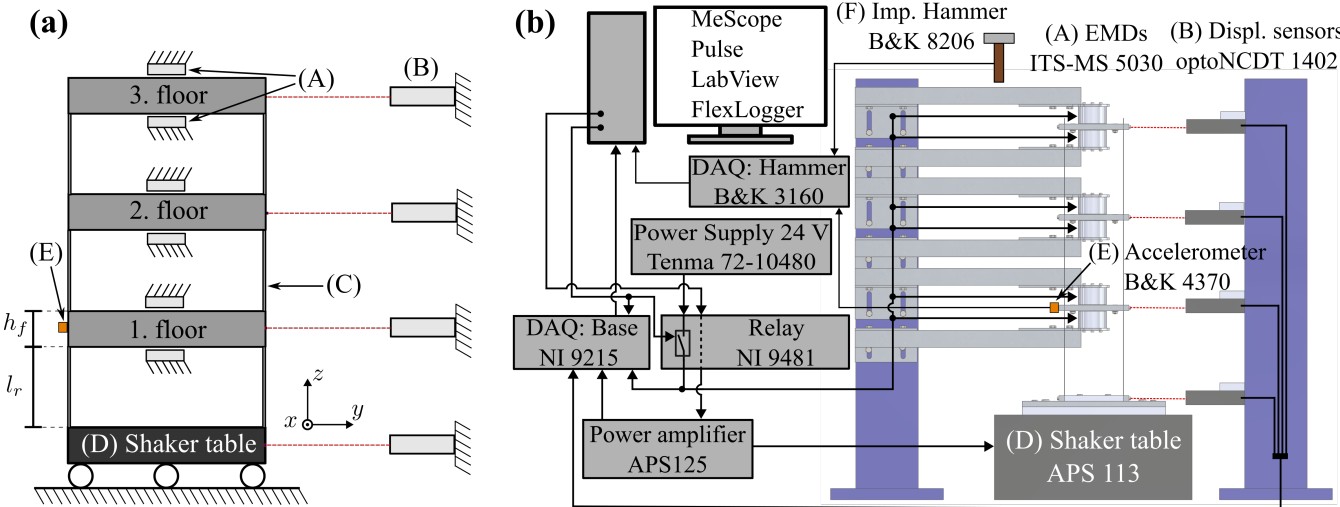

**Figure 1.** Experimental setup. **(a)**: Diagram of SF structure where vibrations primarily occur in the $y$-direction. Each floor has pairs of EMDs (A), and displacement sensors (B). Floors are separated by steel rulers (C). The entire structure is mounted on a shaker table (D). During impact hammer tests only, an accelerometer (E) is attached on the first floor. **(b)**: Instrumentation showing model name for each component

when the EMDs are on (24 V) and off (0 V). From the spectrum, it can be seen that the first three modes have frequencies of

about $(1.8, 5.3, 7.8)$ Hz. The measurements sampled at $50$ Hz are low-pass filtered (stop band frequency of $8.05$ Hz) to prevent aliasing, and subsequently down-sampled to $16.67$ Hz to facilitate faster estimation of the GP-TARMA model, without loosing relevant information about the first three modes. In addition, artificial zero-mean NID noise with RMS level of $1\,\%$ is added to the down-sampled signal.

To validate modal estimates obtained from the output-only base-excitation tests, the modal parameters are also estimated by

standard impact hammer tests (e.g., Ewins (2000); Halvorsen and Brown (1977)). The instrumentation for the impact hammer tests consists of an accelerometer (E) placed on the first floor, an impact hammer with a force transducer, and a B&K 3160 front-end for data acquisition. The test is conducted by impacting each floor multiple times and measuring the resulting acceleration response. The accelerometer and force transducer signals are sampled by the B&K front-end and analysed using B&K Pulse and ME'scope software to estimate modal properties via local FRF curve-fitting. The accelerometer is only mounted on the

first floor during impact hammer tests, thus dismounted prior to base-excitation tests. During hammer tests, the shaker table is fixed. Hammer tests are conducted both with all EMDs on and off, and each test is conducted three times.

## 3.2    Model structure identification

A suitable model structure for modal analysis based on the third-floor response $y(t)$ is identified using the procedure summarized in Table 1. In this controlled experimental setting, the excitation can be considered stationary since the shaker table is

driven by a stationary NID signal. The functional subspace $\mathcal{F}_{MA}$ representing the effect of the excitation therefore only consist of a constant-value bias vector, and the innovations variance is assumed constant (corresponding to $K = N/2$ in Eq. (14)).



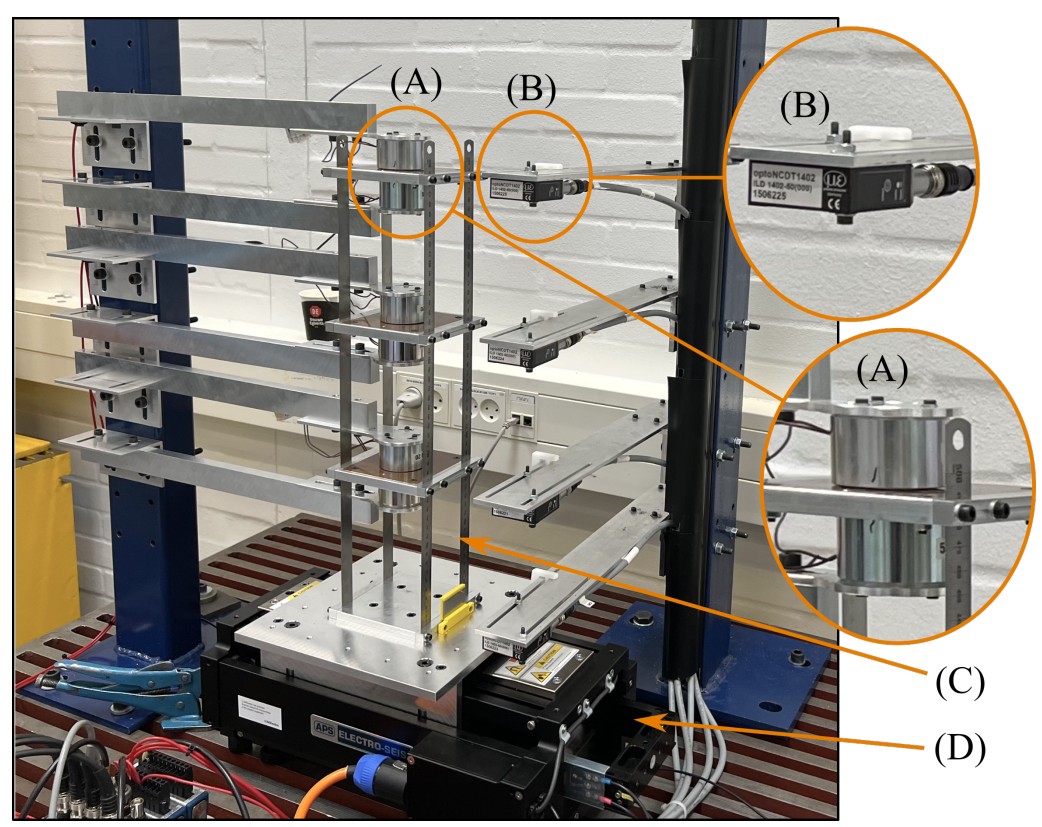

**Figure 2.** Laboratory shear frame test setup (cf. Fig. 1, with same annotations)

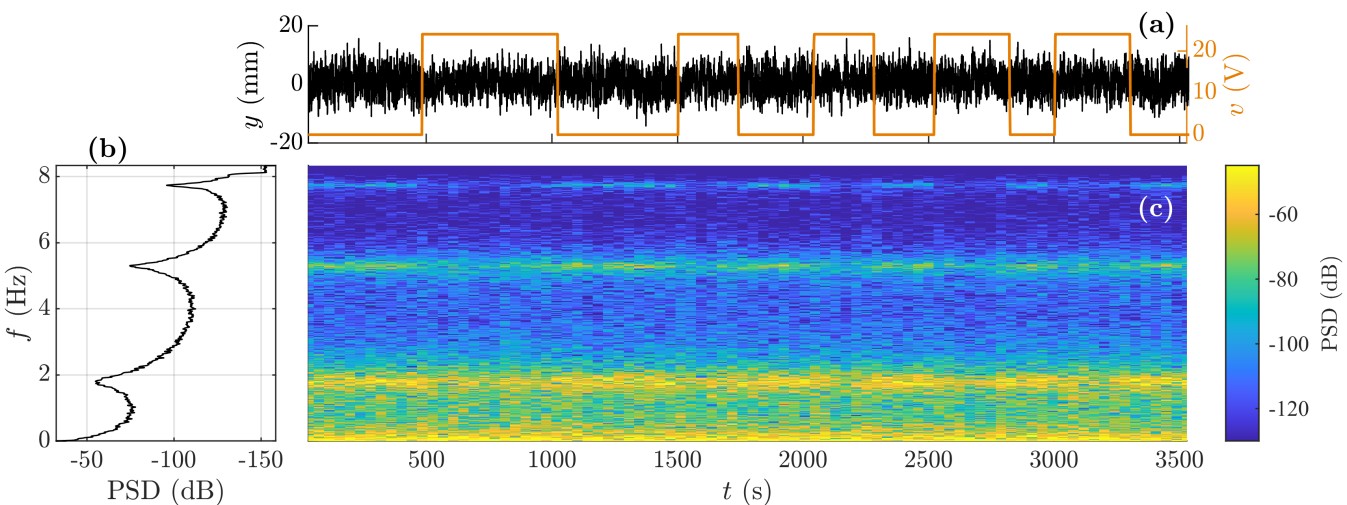

**Figure 3.** Measured response. **(a)**: Displacement of third floor $y(t)$ (—) used as response for analysis, overlaid by measured voltage over electromagnetic dampers $v(t)$ (—) used as EOV; **(b)**: PSD; **(c)**: spectrogram of $y(t)$

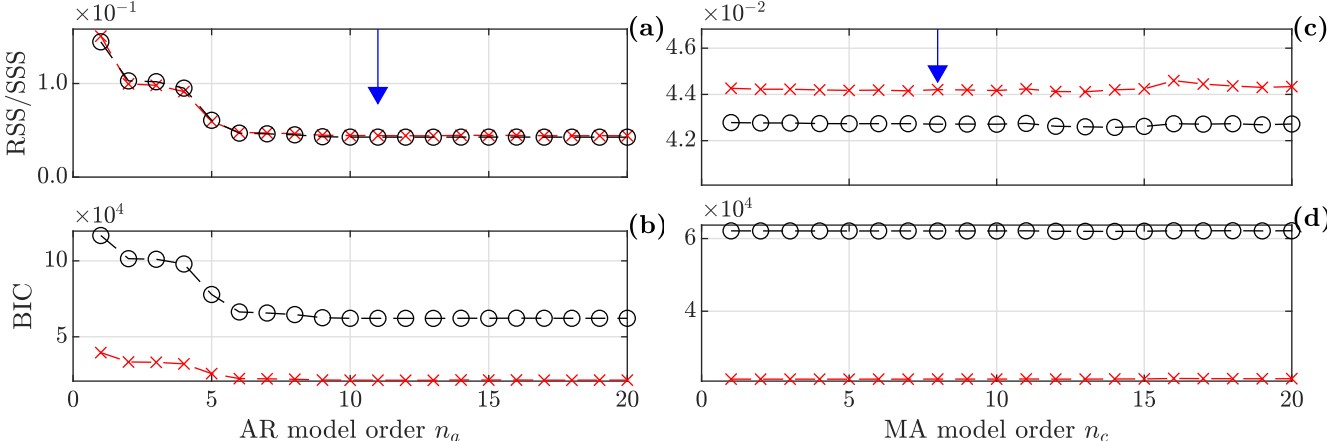

**Figure 4.** Model order selection measures for training/test set (O/X) for varying AR and MA model orders; **(a)** and **(c)**: normalized residual sum of squares; **(b)** and **(d)**: Bayesian Information Criteria. Selected model orders marked by (↓)

To capture the step-like damping variations caused by the EMDs turning on and off, a first order Legendre polynomial in the voltage supplied to magnetic dampers (i.e., the EOV) is included in the functional subspace $\mathcal{F}_{AR}$, along with a constant-valued bias vector to account for the naturally occurring damping. *If* the voltage $v(t)$ had changed gradually (i.e., not binary) from

0 to $24\,\mathrm{V}$, $\mathcal{F}_{AR}$ would likely need to include higher order Legendre polynomials to model the voltage–damping relation. The functional subspaces for the AR- and MA-part are populated as:

$$\mathcal{F}_{AR} = \{\mathbf{A}_0, \mathbf{g}_2(\mathbf{v})\}, \tag{28a}$$

$$\mathcal{F}_{MA} = \{\mathbf{C}_0\}, \tag{28b}$$

where $\mathbf{A}_0 \in \mathbb{R}^{N \times 1}$ and $\mathbf{C}_0 \in \mathbb{R}^{N \times 1}$ are constant-valued bias vectors, $\mathbf{v} \in \mathbb{R}^{N \times 1}$ contains measurements of the voltage over the

EMDs for times $t = 1, \ldots, N$, and $\mathbf{g}_2(\mathbf{v})$ is a first order Legendre polynomial in $\mathbf{v}$. The voltage signal is filtered with a centered moving average filter (better suited for step-signals than regular low-pass filters) with a window width of one second to prevent measurement noise from propagating to the modal parameters (cf. sect. 2.1).

The AR and MA model orders $n_a$ and $n_c$ are selected such that the predictive capabilities of the model in terms of the RSS/SSS and BIC (cf. sect. 2.4) are converged, *and* the model captures modes of interest. In Fig. 4 RSS/SSS and BIC are seen

to converge at about $n_a = 7$ and $n_c = 2$, suggesting the AR and MA model orders should be 7 and 2 or higher. Inspecting the frequency stabilisation diagram in Fig. 5, the three stable poles corresponding to the three natural frequencies of the SF, can be seen to stabilise at model orders $n_a = 11$ and $n_c = 8$, which are selected as the model orders used in downstream analysis. Thus, the driving model selection criteria is the ability to capture the modes of interest, as is typically the case.



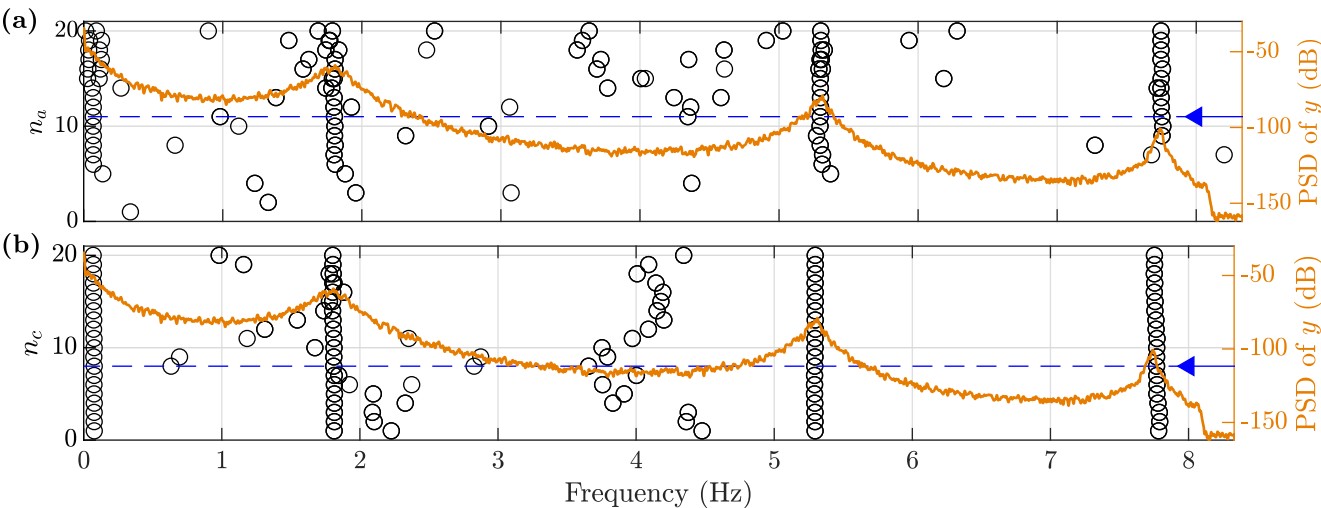

**Figure 5.** Frequency stabilization diagrams showing the time-averaged mean estimates of eigen-frequencies for increasing model orders, overlaid by PSD (——) of response $y$. **(a)/(b)**: AR/MA-order $n_a/n_c$. Selected model orders indicated by (⬅)

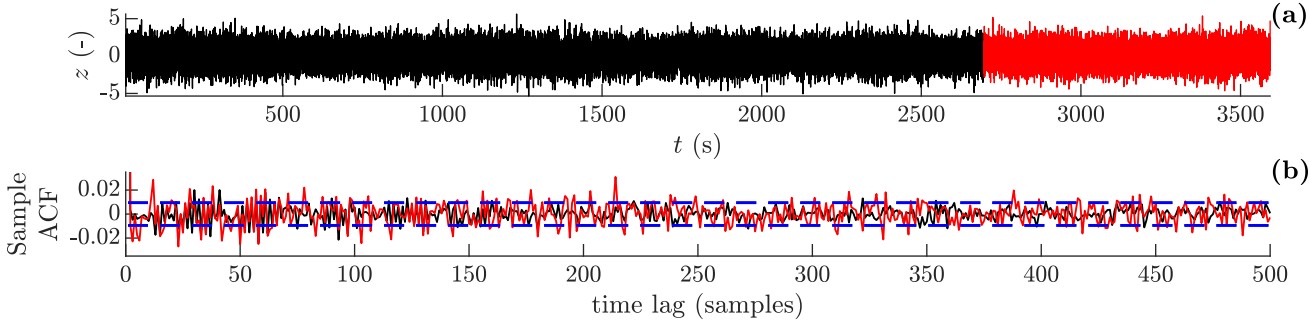

**Figure 6.** Residual analysis from training/test (——/——) set. **(a)**: standardised residuals $z(t)$; **(b)**: ACF of $z(t)$, where the 95 % confidence interval of white noise (- -) is approximated by $\pm 2\sqrt{1/N}$. ACF is by definition unity at lag zero, but not visible in plot

## 3.3 Model validation

In this section the model structure identified in sect. 3.2 is validated using the procedure presented in sect. 2.3, based on the model residuals $e_t$. The standardised residuals and the sample ACF can be seen in Fig. 6. Judging by the time series plot, the standardised residuals appear stationary stochastic, and similar for the training and test set. This observation is supported by the ACF, for which significant correlations only exceed the 95 % confidence limits for white noise at 3 %/4 % of the time lags (i.e., less than 5 %) for the training/test dataset. Furthermore, the sign test shows that the number of sign changes in the residual 395 sequence are well within the 95 % confidence interval of sign changes for a sequence of Gaussian white noise of same length.



**Table 3.** Natural frequency $f_i$ and damping ratio $\zeta_i$ estimates of first to third mode ($i = 1, 2, 3$) for conditions with electromagnetic dampers turned off/on (0/+) estimated with three different methods: GP-TARMA (showing mean estimates and $95\,\%$ *confidence intervals*), SSI estimates averaged for all eight windows (excluding the 30-second estimates), and three repetitions of the hammer test

|  | $f_1$ (Hz) | | $\zeta_1$ (%) | | $f_2$ (Hz) | | $\zeta_2$ (%) | | $f_3$ (Hz) | | $\zeta_3$ (%) | |
| Estimate | 0 | + | 0 | + | 0 | + | 0 | + | 0 | + | 0 | + |
| --- | --- | --- | --- | --- | --- | --- | --- | --- | --- | --- | --- | --- |
| GP-TARMA, mean | 1.81 | 1.82 | 5.9 | 10.6 | 5.29 | 5.29 | 1.0 | 2.7 | 7.76 | 7.79 | 0.6 | 2.1 |
| GP-TARMA, 95% CI | *0.09* | *0.11* | *5.9* | *6.7* | *0.02* | *0.03* | *0.4* | *0.5* | *0.02* | *0.02* | *0.2* | *0.4* |
| SSI (averaged values) | 1.81 | 1.81 | 4.6 | 10.5 | 5.30 | 5.28 | 1.2 | 2.1 | 7.73 | 7.76 | 0.4 | 1.1 |
| Hammer test, test 1 | 1.81 | 1.81 | 6.4 | 10.8 | 5.32 | 5.33 | 1.5 | 3.2 | 7.80 | 7.81 | 0.7 | 1.9 |
| Hammer test, test 2 | 1.82 | 1.82 | 6.5 | 10.5 | 5.32 | 5.33 | 1.5 | 3.1 | 7.80 | 7.81 | 0.6 | 1.8 |
| Hammer test, test 3 | 1.82 | 1.81 | 6.1 | 10.9 | 5.32 | 5.32 | 1.5 | 3.4 | 7.79 | 7.79 | 0.7 | 2.1 |

The above whiteness tests, based on both standardised residuals and sign changes, suggests that the present GP-TARMA model is valid and well-suited for downstream analysis.

### 3.4 Results: Modal parameter estimates

In this section modal parameter estimates computed from the validated GP-TARMA model of the third-floor displacement

response $y(t)$, along with modal parameter estimates based on hammer test are presented. The GP-TARMA model is estimated using 2700 s of data, corresponding to 44820 samples (at sampling frequency of 16.67 Hz), and about 4860, 14310, and 21060 oscillations of first to third mode. Figure 7 shows predicted response against measured, GP-TARMA estimates of natural frequencies and damping ratios with $95\,\%$ confidence intervals, and SSI estimates for comparison. The SSI algorithm used is correlation-driven (Peeters and De Roeck, 1999; Brincker and Ventura, 2015), and the stable poles corresponding to the three

modes are selected manually from frequency stabilization diagrams. The SSI estimates are also based on the displacement response of the third floor $y$, and computed in windows of constant voltage over the electromagnetic dampers (eight windows seen in plot). In addition, SSI estimates based on measurements in 30 second windows are shown for the sixth and seventh window (2043 to 2523 s).

The predicted response in Fig. 7 is observed to model the measured response very well. The voltage over the electromagnetic

dampers $v(t)$ is overlaid the plot, and zero volt means no added damping from the electromagnetic dampers. A slight difference in response levels can be seen by comparing sections of the response with and without the electromagnetic dampers activated.

Good agreement for natural frequency estimates between GP-TARMA and SSI is observed. The modal parameters estimates are also summarized in Table 3. Furthermore, the widest confidence intervals for the GP-TARMA natural frequency estimates are $\pm(0.11, 0.03, 0.02)$ Hz for first, second, and third mode seem representative for the spread of the SSI estimates, as seen in

Table 3.



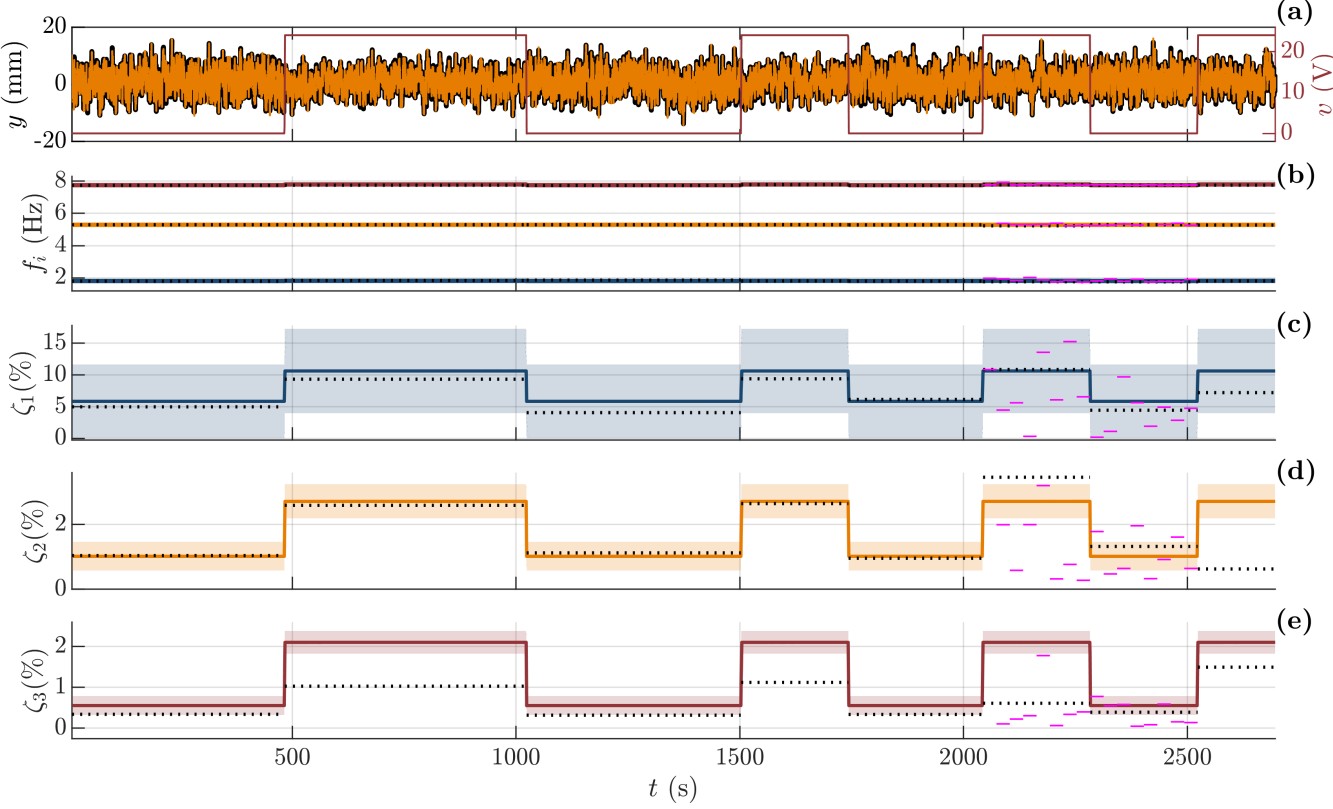

**Figure 7.** GP-TARMA estimates compared to SSI estimates. **(a)**: measured/predicted (━/━) response $y$ overlaid by voltage $v$ over EMDs (━); **(b)**–**(e)**: GP-TARMA modal parameter estimates for first/second/third (━/━/━) mode, where mean estimates shown by solid lines, shaded area indicate estimated $95\,\%$ confidence intervals. SSI estimates for constant-voltage windows (⋯), and for 30-second estimates (━). **(b)**: natural frequencies $f_i$; **(c)**–**(e)**: damping ratios $\zeta_i$ of first to third mode

The figure (and table) shows a good agreement between SSI and GP-TARMA damping estimates, where the best agreement is observed for the second mode. The recommended minimum measurement time (Brincker and Ventura, 2015) is dictated by the third mode (with EMDs off) and is about $T_{min} = \frac{10}{f_3 \zeta_3} = 214\,\text{s}$ (based on the hammer test estimates in Table 3), i.e., the eight constant-voltage windows of minimum $240\,\text{s}$ should be sufficiently long for adequate SSI estimates. The GP-TARMA damping ratio estimates in Fig. 7 illustrates the idea of letting the AR coefficients, and consequently the modal parameters, be represented by EOV-dependent basis functions. In this case, the voltage over the EMDs dictates the instantaneous changes in the damping estimates. In comparison, the 30-second SSI estimates (2043 to 2523 s) are seen to be inconsistent, and many deviate considerably from the other estimates. This illustrates the need for dedicated methods for short-term damping estimation, such as the GP-TARMA approach. Using the GP-TARMA approach comes with the cost of specifying basis functions capable of representing (i.e., correlating with) the underlying causes of nonstationarity in the response, for which prior knowledge of the

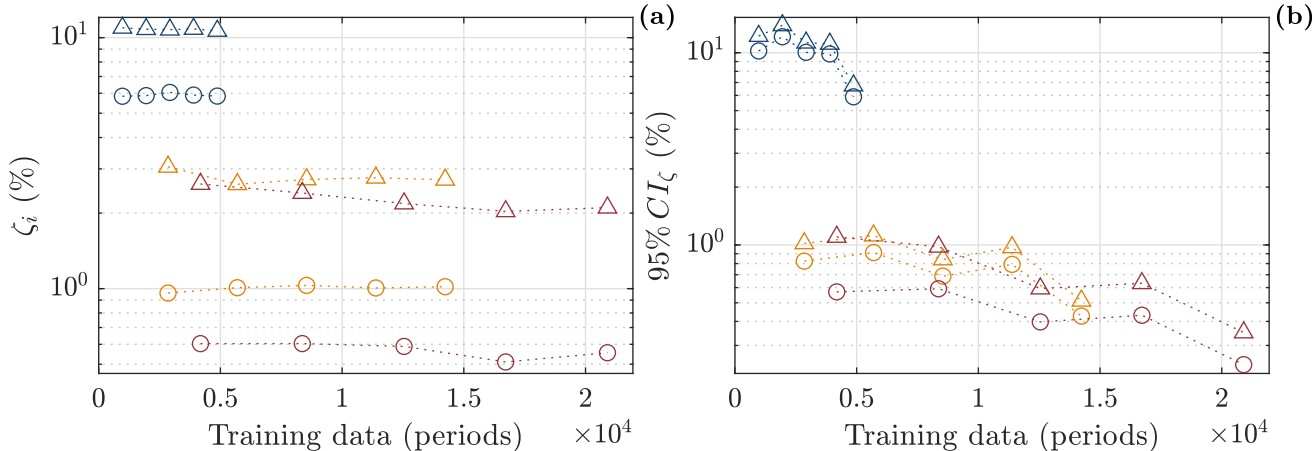

**Figure 8.** Convergence of damping estimates with respect to training data. **(a)**: Mean estimates; **(b)**: 95 % confidence interval. Modes indicated by color (first/second/third), estimates for electromagnetic dampers off/on (O/△)

system is helpful. That is not needed for standard OMA methods such as SSI, although the validity of the LTI assumptions must be examined.

The widest damping estimate confidence intervals for first to third mode are $\pm(6.7, 0.5, 0.4)$ %, i.e., the damping estimate for the first mode is associated with the highest uncertainty of the three modes, as for the natural frequencies. This might

reflect that the training data contains more oscillation periods of the higher modes. To help elucidate this hypothesis, Fig. 8 shows the convergence of mean damping ratio estimates and corresponding confidence intervals for all three modes in terms of oscillation periods. The results seen in Fig. 7 correspond to the estimates based on the largest set of training data in Fig. 8. The figure indicates that only the third mode might have converged in terms of the confidence intervals, and especially the first mode would seem to benefit from being estimated from more data. However, the mean estimates do not change much with the

increase in training data. This suggests that the mean estimates might be representative, despite the corresponding confidence intervals are wide.

Table 3 shows, that the GP-TARMA and SSI natural frequency estimates agree well with the estimates from the three hammer tests. As for the damping ratios, the three hammer test estimates compare well to the GP-TARMA and SSI estimates, in the sense that the are well within the same order of magnitude. The hammer test estimates cannot be expected to agree

completely with GP-TARMA or SSI estimates, because the estimates are based on data from two fundamentally different experimental tests in terms of excitation (impulse vs. stochastic), amount of energy input, temporal and spatial distribution of energy input, etc.

The SF test results presented in this section experimentally validates the efficacy of the GP-TARMA method for providing representative short-term damping estimates, and illustrates its efficacy in the short-term case in comparison to a traditional

OMA method.



## 4 Full-scale wind turbine test: Instability experiment

In this section the GP-TARMA approach is applied to edgewise blade response measurements of a 7 MW wind turbine with a rotor diameter of $154\,\mathrm{m}$. Specifically, the method is used to estimate short-term, linear equivalent modal damping of edgewise rotor modes, which are deliberately driven to flutter-like instabilities, corresponding to negative damping values.

The experiments were conducted with an SWT-7.0-154 prototype wind turbine located at the DTU Wind Test Centre in Østerild, Denmark. The measurements collected in December 2018 were published by Volk et al. (2020), following up an instability field validation study (Kallesøe and Kragh, 2016). The edgewise blade modes were driven to flutter-like instabilities by momentarily allowing the rotor to run $10\,\%$ above the expected stability limit, by changing the wind turbine controller parameters. As in Volk et al. (2020), the response used for the analysis is an edgewise blade bending response, obtained using

fiber Bragg optical strain sensor positioned $72\,\mathrm{m}$ outboard on the blade (i.e., close to the blade tip), measuring the bending response in the rotating frame of reference and is in sect. 4 referred to as $y(t)$. All frequencies are normalized by the first edgewise blade frequency, and rotor speeds are normalized by the critical rotor speed of the first edgewise mode as in Volk et al. (2020).

Figure 9 shows the time series, spectrum, and spectrogram of the measured edgewise blade response $y(t)$ along with nor-

malized rotor speed. The figure shows a clear relation between high rotor speeds and occurrence of "unstable" (high response levels) modes, which appear at normalized frequencies of 1.0 and 2.5. Strong response of these modes is especially prevalent at 150–230 s and 340–420 s.

The amount of available training data from this test is quite small relative to the model complexity (i.e., number of parameters to be estimated) needed for the GP-TARMA model to represent the complex response and underlying system. The bandwidth

of $y(t)$ has therefore been reduced compared to the measurement data, and artificial zero-mean NID noise with a $5\%$ RMS of $y(t)$ is added to the response $y(t)$. These steps been taken to use the limited measurements as efficiently as possible, and to avoid over-fitting (cf. sect. 2.6).

### 4.1 Model structure identification

The model structure is identified following the procedure in Table 1. The functional subspace $\mathcal{F}_{AR}$ used to represent the

AR-coefficients consist of a first order Legendre polynomial in the rotor speed $\Omega(t)$ along with a constant-valued bias vector. This allows the model to capture the effect of the varying rotor speed on the modal damping, and the presumably time-invariant contribution from the structural damping. The MA-coefficients are represented by harmonic functions of the azimuth angle $\psi(t)$, to account for the nonstationary 1P effect from the changing direction of gravity in the rotating frame. Thus, the functional subspaces for the AR- and MA-part are defined as:

$$\mathcal{F}_{AR} = \{\mathbf{A}_0, \mathbf{g}_2(\mathbf{\Omega})\}, \tag{29a}$$

$$\mathcal{F}_{MA} = \{\mathbf{C}_0, \mathbf{h}_2(\boldsymbol{\psi}), \mathbf{h}_3(\boldsymbol{\psi})\}, \tag{29b}$$





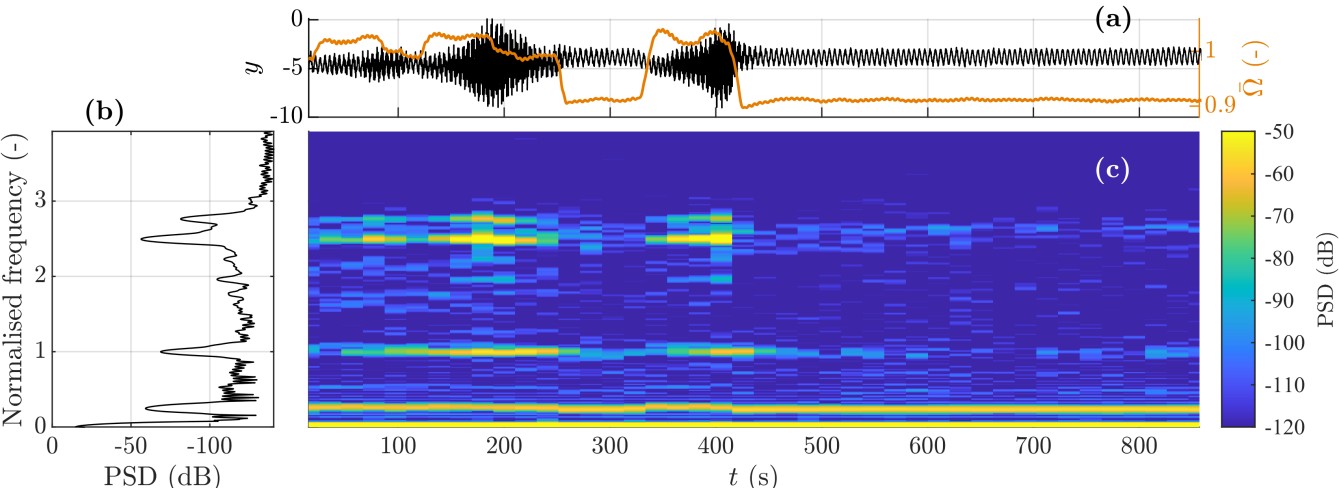

**Figure 9.** Measured response. **(a)**: Edgewise bending response $y$ (━), overlaid by *normalized* rotor speed $\bar{\Omega}$ (━); **(b)**: PSD; **(c)**: spectrogram of edgewise bending response $y(t)$

where $\mathbf{A}_0 \in \mathbb{R}^{N \times 1}$ and $\mathbf{C}_0 \in \mathbb{R}^{N \times 1}$ are real-valued bias vectors, $\mathbf{\Omega} \in \mathbb{R}^{N \times 1}$ and $\boldsymbol{\psi} \in \mathbb{R}^{N \times 1}$ contains rotor speed and azimuth angle measurements for $t = 1, \ldots, N$, $\mathbf{g}_2(\mathbf{\Omega})$ is a first order Legendre polynomial in the rotor speed $\Omega$, and $\mathbf{h}_2(\boldsymbol{\psi}) = \sin(\boldsymbol{\psi})$ and $\mathbf{h}_3(\boldsymbol{\psi}) = \cos(\boldsymbol{\psi})$ are the harmonic functions in the azimuth angle $\boldsymbol{\psi}$. Thus, the EOVs used in the model are the rotor speed

and azimuth angle, $\boldsymbol{\xi} = [\mathbf{\Omega}, \boldsymbol{\psi}] \in \mathbb{R}^{N \times 2}$. Both EOVs are zero-phase low-pass filtered with a cut-off frequency of $0.3 \,\text{Hz}$ (cf. sect. 2.6), and the innovations variance is estimated using Eq. (14) with a window length of $3.33 \,\text{s}$, i.e., the GP-TARMA model can estimate damping (and natural frequency) variations down to 3.33 seconds.

The AR (MA) model orders $n_a$ ($n_c$) are selected based on the predictive capability (minimizing BIC and RSS/SSS) and ability to capture the modes of interest (cf. sect. 2.4), which in this case have normalized frequencies of about 1 and 2.5. As in

sect. 3.2, the model structure is dictated by the ability to capture the modes of interest, i.e., the predictive capability converges at a lower model complexity. Figure 10 shows stabilization diagrams with respect to AR and MA orders. The selected model orders are $n_a = 17$ and $n_c = 5$, as these are the lowest model orders at which the poles can be considered stable. RSS/SSS and BIC converge at $n_a = 5$ and $n_c = 4$ (plots not included).

### 4.2   Model validation

The model structure identified in sect. 4.1 is validated in this section, based on analysing the model innovations $e_t$ as presented in sect. 2.3. Figure 11 shows the time series, estimated ACF and spectrum of the standardised residuals of the estimated GP-TARMA model with model structure identified in sect. 4.1. The time series of the standardised residuals does not appear as stationary white noise, since the amplitude (i.e., variance) exhibits time-varying behaviour especially at $200 \,\text{s}$ and $400 \,\text{s}$. This is because the estimated innovations variance $\hat{\sigma}^2_{e,t}$ is larger at those instances, which are the times of the measured instabilities.

The ACF also indicates that the standardised residuals do not resemble white noise, as the ACF for both the training and



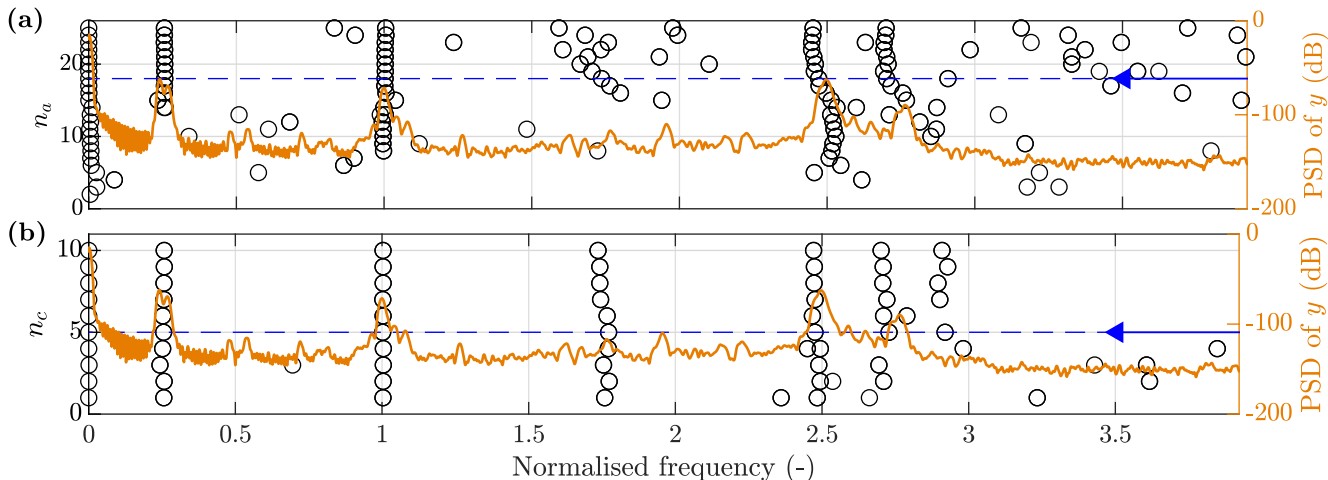

**Figure 10.** Frequency stabilization diagrams showing the time-averaged mean estimates of eigen-frequencies for increasing model orders, overlaid by PSD (━) of response $y$. **(a)/(b)**: AR/MA-order $n_a/n_c$. Selected model orders marked by (←)

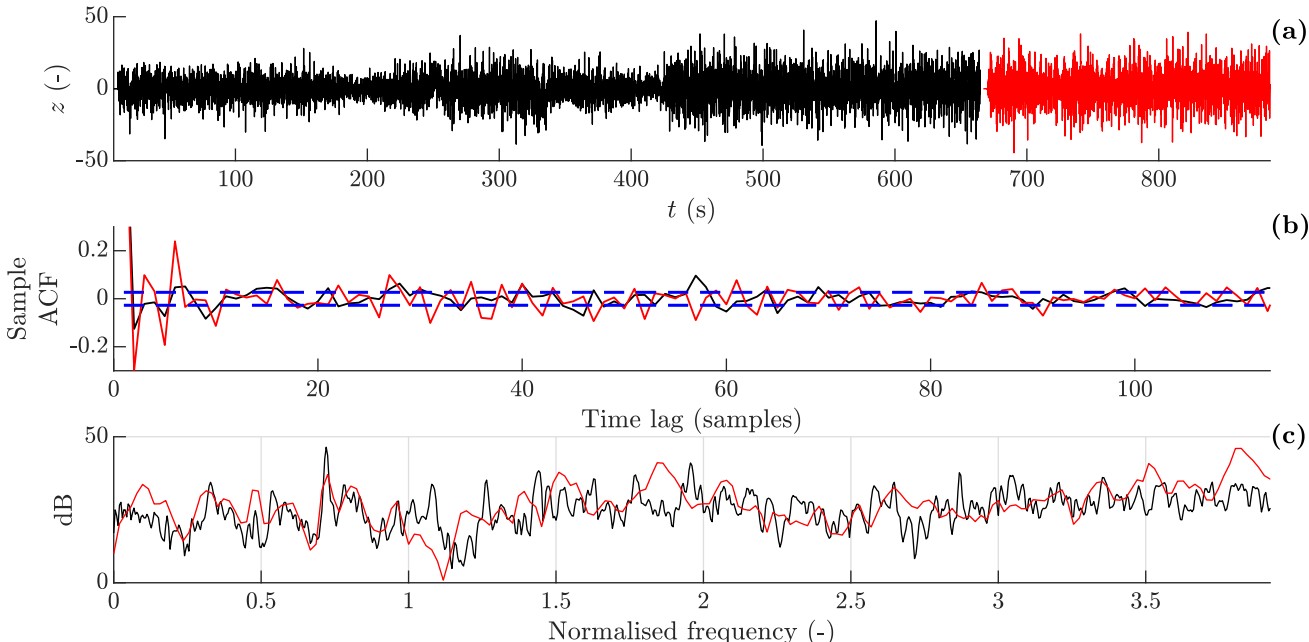

**Figure 11.** Residual analysis from training/test (━/━) set. **(a)**: standardised residuals $z(t)$; **(b)**: ACF of $z(t)$, where the 95 % confidence interval of white noise (- -) is approximated by $\pm 2\sqrt{1/N}$. ACF is by definition unity at lag zero, but not visible in plot; **(c)**: spectrum of $z(t)$

test data exceeds the 95 % confidence level for white noise at 13 % of the time lags. The spectrum shows a distinct peak at normalized frequency of 0.72, which coincides with three times the (average) rotor speed, indicating that the model does not



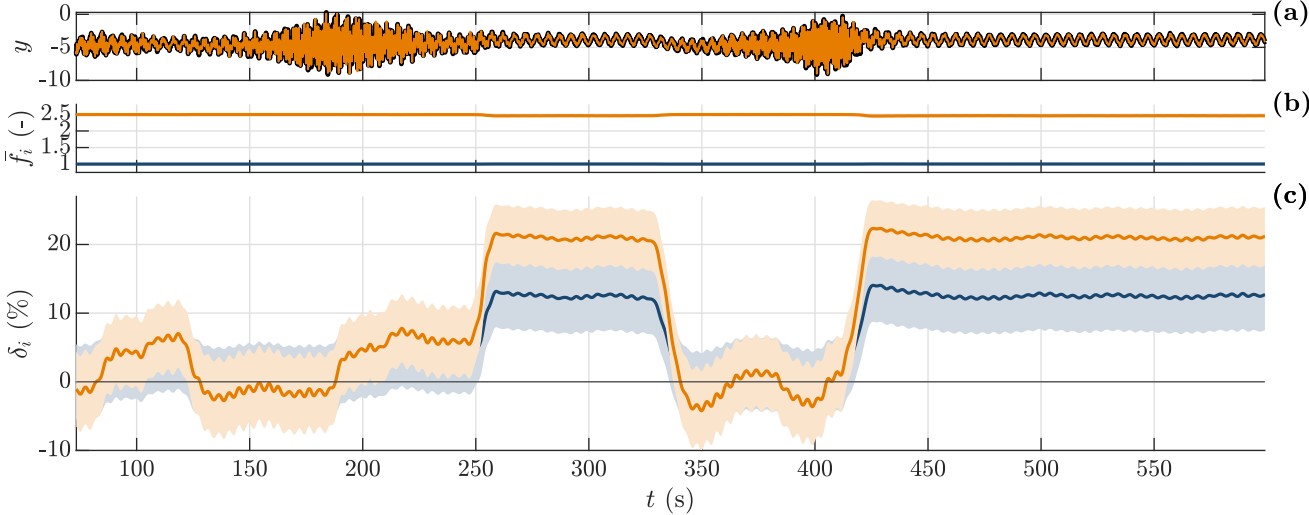

**Figure 12.** GP-TARMA model estimates as function of time. **(a)**: measured/predicted (━/━) edgewise blade response $y$; **(b)** and **(c)**: Modal parameter estimates for first/second (━/━) backward whirling modes, solid lines depict mean estimates and shaded area indicate 95 % confidence intervals. **(b)**: normalized natural frequencies $\bar{f}_i$; **(c)**: damping (logdec) $\delta_i$

adequately capture the 3P effect. However, the 3P frequency is well separated from the frequencies of the modes of interest, so it is not likely to affect modal parameter estimates. In unison with the other whiteness measures, the number of sign changes
in the residual sequence is not within the 95 % confidence intervals of the number of sign changes for an ideal white noise sequence, supporting that the model residuals do not resemble white noise.

The residual analysis implies that the NID assumption of the innovations is violated, i.e., the model does not completely represent the statistical structure of the response. However, the frequencies at which the residuals have the largest magnitudes do not coincide with the frequencies of the two modes of interest. Thus, the modal parameters computed from the GP-TARMA
based on the present dataset is not expected to be entirely accurate but may still offer some insight. More available measurement data (ideally including more instability measurements) for model training could improve the model accuracy by enabling more robust and accurate model parameter estimates. In addition, a single-output model (as the actual GP-TARMA model) cannot account for the whirling effect, since the frequencies of the forward and backward whirling modes coincide in a response measured in the rotating frame. Such model-form error might cause correlated model residuals.

**4.3 Results: Modal parameter estimates**

Figure 12 shows the predicted response of the GP-TARMA model, and the corresponding modal parameter estimates with uncertainties. For direct comparison to Volk et al. (2020), the damping estimates are reported in terms of *logarithmic decrement* (logdec) $\delta$, which (for small damping) is related to *damping ratio* $\zeta$ by $\delta = 2\pi\zeta$. The estimated frequencies correspond well with peaks in magnitude seen in the spectrum and spectrogram of response $y$ in Fig. 9, and the average 95 % confidence intervals

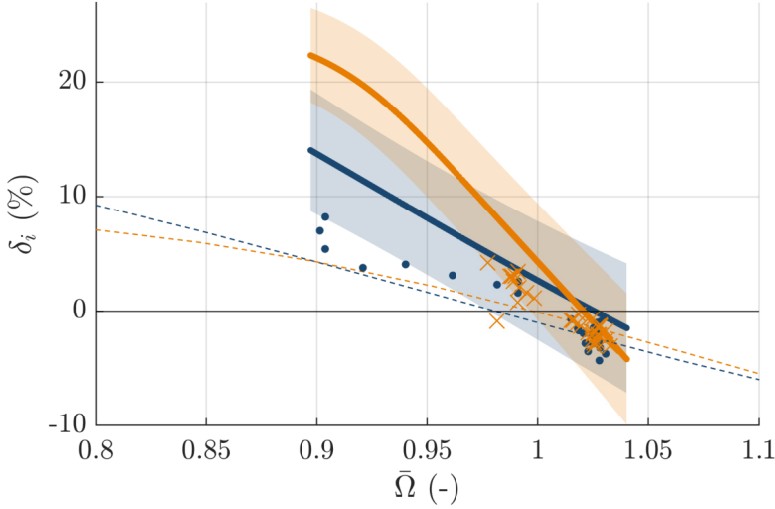

**Figure 13.** Modal damping of first and second backward whirling mode in terms of logarithmic decrement $\delta_i$ as function of normalized rotor speed $\bar{\Omega}$. First/second backward whirling mode: GP-TARMA (experiment) (━/━), HAWCStab2 blade-only (Volk et al., 2020) (- -/- -), exponential fits (experiment) (Volk et al., 2020) (●/✕)

for the first and second mode are $\pm(0.01, 0.02)$ Hz. Both edgewise modes can be seen to become "unstable" (i.e., negatively damped), and most noticeable between 128–187 s and 341–405 s. The time instants of the mean damping estimates crossing zero damping coincides well with the amplitude enveloped changing from exponentially increasing or decreasing or vice versa, i.e., the mean damping estimates are qualitatively meaningful. The average 95 % confidence intervals of the damping estimates for the first and second more are $\pm(5.2, 4.8)$ % logdec, corresponding to about $\pm(0.8, 0.8)$ % in terms of critical damping.

520       The uncertainty associated with damping estimates may seem considerable, as the confidence intervals cover both positive and negative damping values during the flutter-like instabilities. The uncertainty might be reduced by addressing the two limiting factors alluded to in sect. 4.2, namely the sparse training data, and the inability of the single-output GP-TARMA model to represent the whirling effect.

          Figure 13 shows the damping estimates in Fig. 12 as function of normalized rotor speed rather than time, which enables
assessment of stability limits for the identified modes. The GP-TARMA estimates are plotted against two sets of damping estimates from Volk et al. (2020), consisting of experimental estimates (based on the same response data) and predictions computed using HAWCStab2 (Hansen et al., 2018) based on a numerical model of the tested wind turbine. The experimental damping estimates are obtained using logarithmic decrement fits (i.e., exponential fits of response envelope) of bandpass filtered (near each resonance frequency) response signals. The exponential fits are performed on four sections of the data; two
sections with exponentially increasing (negative damping) amplitude and two with exponentially decreasing (positive damping) amplitude (see Volk et al. (2020) for details).





**Table 4.** Comparison of stability limit estimates of first and second edgewise backward whirling mode in terms of normalized rotor speed $\bar{\Omega}$

| Mode | Experiment: GP-TARMA | Experiment: Exponential fits (Volk et al., 2020) | HAWCStab2 blade-only (Volk et al., 2020) |
|---|---|---|---|
| First edgewise backward whirl | 1.03 | 1.00 | 0.98 |
| Second edgewise backward whirl | 1.02 | 1.01 | 1.00 |

The figure shows that the GP-TARMA estimates agree quite well with the exponential fit estimates, and especially near the critical rotor speeds, but the GP-TARMA damping estimates tend to be higher than the exponential fit estimates. Table 4 shows the critical rotor speed estimated by each of the three approaches. In terms of the critical rotor speed, the HAWCStab2 predictions are slightly more conservative in comparison to the two experimental results. However, comparing the $\delta_i(\bar{\Omega})$ curve slopes in Fig. 13, the HAWCStab2 results predict less rotor speed sensitivity, i.e., are less conservative with respect to how quickly the instabilities occur. But these discrepancies are small relative to the GP-TARMA damping estimate uncertainties, and the unquantified (but inevitable) uncertainties of the exponential fit estimates.

## 5   Conclusions

A recently proposed approach based on a Gaussian Process Time-dependent Auto-Regressive Moving Average (GP-TARMA) model for short-term damping (and natural frequency) estimation from output-only vibration response measurements for vibrating structures influenced by environmental and operational variability has been experimentally tested and validated with two distinctly different experimental setups: a laboratory shear frame structure with time-varying damping properties achieved with electromagnetic dampers, and a full-scale 7 MW wind turbine prototype which was deliberately driven to flutter-like instabilities. The primary idea of the GP-TARMA approach is to condition the model parameters on measured time series of environmental and operational variables, which may enable short-term tracking of system parameters like time-varying damping and natural frequencies.

An experimental setup consisting of a shear frame structure equipped with electromagnetic dampers was presented and shown to effectively realize a system with abruptly changing damping. Short-term natural frequencies and damping ratios were estimated using the GP-TARMA model and shown to compare well to SSI and hammer test estimates in cases where the system was time-invariant. Uncertainties were observed to be larger for first mode compared to second and third, but this could be explained by the first mode being trained on effectively less training data. GP-TARMA damping estimates was compared to short-term SSI estimates based on windows of 30 s measurements. The short-term SSI estimates were observed to be inconsistent and deviating from the remaining estimates, which illustrated the effectiveness of the GP-TARMA method for short-term damping estimation relative to traditional OMA methods. The laboratory test validated the efficacy of the GP-TARMA approach for short-term damping (and natural frequency) estimation, given a sufficient amount of training data and representative model structure.





The GP-TARMA model was also tested using edgewise blade deflection measurements from a full-scale 7 MW wind turbine prototype during a flutter test. First and second edgewise backward whirling mode were found to exhibit flutter-like instabilities, in agreement with a previous study. The mean damping estimates were considered qualitatively meaningful, as the GP-TARMA model predicted negative damping for two modes coinciding in time and frequency with exponentially increasing vibration amplitude. The mean damping estimates also compared quite well with estimates from a previous study obtained from the same data. The estimated stability limits, i.e., the rotor speeds at which the damping becomes zero, showed quite good agreement with a previous study. However, the model validation implied that the model residuals did not resemble white noise, meaning that the GP-TARMA model trained on the available data cannot be expected to be entirely accurate. The correlated model residuals and uncertainties of the damping estimates could potentially be reduced by training the GP-TARMA model on more data and extending the GP-TARMA model to a multiple-output model to better represent whirling modes.

The GP-TARMA approach appears an effective way of estimating short-term damping based on output-only measurements, provided enough training data and a representative model structure. The use of GP-TARMA models for analysis of transient instabilities has been showcased. SSI and other standard OMA methods are easier to implement and apply than the GP-TARMA approach since it does not require much prior knowledge of the system, i.e., these should be used for applications where the LTI assumptions are valid but may be inadequate for applications with considerable short-term EOC variability.

*Code and data availability.* Laboratory measurements are available upon request, wind turbine test data is confidential. Code might be shared upon request.

*Author contributions.* Conceptualization and methodology: KLE, PC, and JJT; validation: KLE, PC, LMS, and JJT; investigation and visualization: KLE and LMS; writing (original draft): KLE and LMS; supervision and writing (review and editing): PC and JJT; software, formal analysis: KLE

*Competing interests.* The authors declare that they have no known competing financial interests or personal relationships that could have appeared to influence the work reported in this paper.

*Acknowledgements.* This work is partially funded by Innovation Fund Denmark (grant 0153-00179B). Shaker table used in laboratory setup is donated by COWIfonden (grant A-155.01).



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
