# Peer review of "Experimental validation of a short-term damping estimation method for wind turbines in nonstationary operating conditions"

_Wind Energy Science, 2023_

## Author Response (AR1)

**Author response to reviewer comments**: WES-2023-168 *Experimental validation of a short-term damping estimation method for wind turbines in nonstationary operating conditions* by Kristian Ladefoged Ebbehøj, Philippe Jacques Couturier, Lars Morten Sørensen, and Jon Juel Thomsen.

We would like to thank the reviewers for their time and consideration, and for the constructive comments and suggestions. The comments are addressed in the revised manuscript, as can be seen below. Changes are highlighted as added/deleted, and references to numbered sections, figures, etc. is with respect to the revised manuscript.

**Referee #1**

**Comment 1.0** (paper summary omitted): *The paper clearly describes the proposed methodology and demonstrates, through experimental analyses, the validity of the proposed method for identifying short-term damping in time-varying vibrating structures under nonstationary operating conditions. I therefore advise the editor to accept the paper after a minor revision.*

With the intent to improve the quality of the paper, I would like to recommend the authors the following modifications:

Response: Thank you for your general, positive remark.

**Comment 1.1**: The paper should briefly discuss the limitations of the proposed methodology, e.g. problems that might occur in the presence of strongly nonlinear properties in the considered structure.

**Response:** We have added a discussion on the limitations in the revised manuscript (Section 1, 9th paragraph):

"The capabilities of the GP-TARMA model to track EOC variability is limited by the extent of how well the basis functions capture the nonstationary of the response (e.g., slow or fast variations), and fundamentally by the measurement sampling rate. While the GP-TARMA model may be nonlinear with respect to EOVs, it is linear with respect to the response it is modelling, i.e., representing an equation of motion that is linear in the dependent variables. Consequently, the model cannot capture strongly nonlinear system properties. However, weakly nonlinear effects on the effective natural frequencies and damping ratios may be approximated if these nonlinear effects correlate with operational states represented by the EOV-dependent basis functions."

**Comment 1.2**: In section 3.2 the model orders (Nc and Na) of the ARMA model are selected for the three-storey shear frame using the procedure outlined in section 2.4 and Table 1. Figure 4 shows the trend of the BIC and RSS/SSS indices for different model orders, nonetheless, it is not completely clear how this figure is used to identify the convergence of the model order adopting the proposed procedure of section 2.4. The authors should improve the description of this step in section 3.2. **Response:** Thank you for pointing to this. We have clarified the use of the BIC and RSS/SSS plots in the revised manuscript as follows:

• Sect. 2.4, 2nd paragraph: "Typically, the predictive performance converges at lower model orders <del>compared to</del> than the model orders required to capture the modes of interest<del>, i.e., the latter is typically the driving selection criteria</del>. This means the convergence of the predictive performance is typically a necessary but insufficient condition."

• In the response to Comment 1.3: A reference to sect. 2.4. has been added to, as

• Sect. 3.2,  $2^{nd}$  paragraph: "The AR and MA model orders  $n_a$  and  $n_c$  are selected such that the predictive capabilities of the model in terms of the RSS/SSS and BIC (cf. sect. 2.4) are converged, *and* the model captures modes of interest. In Fig. 4 RSS/SSS and BIC are seen to converge at about  $n_a = 7$  and  $n_c = 2$ , suggesting the AR and MA model orders should be 7 and 2 or higher. constituting lower boundaries for the AR and MA model orders. Next step is to assess the model orders required to capture the modes of interest."

**Comment 1.3**: Figure 5 shows the stabilization diagram for the three-storey shear frame: despite the convergence of RSS/SSS and BIC indices occurs at a model order equal to Nc = 2and the stabilization diagram seems to show stable poles for all the considered modes at Nc= 3 / Nc = 4, a model order Nc = 8 has been used in the final ARMA model. The authors should justify this choice, highlighting their motivations.

**Response:** We appreciate you have drawn our attention to this, and have clarified the reasons for this choice in the revised manuscript:

"Inspecting the frequency stabiliszation diagram in Fig. 5, the three stable poles corresponding to the three natural frequencies of the SF7 can be seen to be stabiliszed at model orders  $n_a = 11$  and  $n_c = 8$ , which are selected as the model orders used in downstream analysis. For the MA model order, the poles arguably converge already at  $n_c = 4$ , but the frequency of the third mode changes slightly until stabilizing at  $n_c = 8$ . This minor consideration can be taken because the amount of training data is large relative to the model complexity, i.e., the risk of over-fitting is small. Thus, In this case the driving model selection criteriaon is the ability to capture the modes of interest, as is typically the case (cf. sect. 2.4)."

**Referee #2**

**Comment 2.0** (paper summary omitted): In the reviewer's opinion, the manuscript is well organized and well written. Publication is recommended before minor revisions addressing the following issues:

**Response:** Thank you for positive general comment.

**Comment 2.1**: According to the authors' comments on Figure 11 in Section 4.2, the time series of the model residuals is not a stationary white noise. Consequently, the authors state

that "The residual analysis implies that the NID assumption of the innovations is violated, i.e., the model does not completely represent the statistical structure of the response". Does it mean that the time series of the model residuals shall always be a stationary white noise for the GP-TARMA model to be accurate? This seems to contradict Section 2.3, where the GP-TARMA model residuals can be non-stationary and a "whiteness" test of the model residuals is possible also in the non-stationary case (see Eq.(16)). Please clarify.

**Response:** Thank you for pointing this out. We have clarified the distinction between standardized and non-standardized residuals in the revised manuscript as follows:

- Sect. 2.3, above Eq. (15): "An approach to partially circumvent this issue is to normalize standardize the residuals with using the estimated time-varying innovations variance δ2e,t (Fouskitakis and Fassois, 2002):"
- Sect 2.3, below Eq. (15): "Given that  $e_t$  is a zero-mean white noise sequence with time-varying variance that is adequately approximated by  $\hat{\sigma}_{e,t}^2$ , the standardized residuals  $z_t$  are stationary. For nonstationary residuals, the ACF test is sensitive to the accuracy of the time-varying innovations variance estimate, and should thus be supplemented by a test insensitive to the innovations variance estimate. Using the standardized residuals  $z_t$  for the ACF test renders the test sensitive to the accuracy of the time-varying innovations variance estimate. It should thus be supplemented by a test insensitive to such an estimate."
- Sect. 4.2, 1st paragraph: "The model structure identified in sect. 4.1 is validated in this section, based on analyszing the model innovations residuals  $e_t$  ( $t = 1 + n_m, ..., N$ ) as presented in sect. 2.3. Figure 11 shows the time series, estimated ACF and spectrum of the standardised standardized residuals  $z_t$  of the estimated GP-TARMA model with model structure identified in sect. 4.1."
  - These changes are also implemented in sect. 3.3, 1st paragraph: "(...) based on the model residuals  $e_t$  ( $t = 1 + n_m, ..., N$ ). The standardised standardized residuals  $z_t$  and the sample ACF (...)"

**Comment 2.2**: *A model validation strategy discussed in Section 2.3 is the "cross-validation". Can the authors apply it to the wind turbine for double-checking the results of the residual analysis?**

**Response:** Thank you for raising this question. We have clarified why cross-validation may not be suitable for this case in the revised manuscript in sect. 4.2, 1st paragraph (interpreting Fig. 11):

- "The time series of the standardized residuals for the training data does not appear as stationary white noise, (...)"
- "Comparing the standardized residuals for the test data to those of the training data, the time-varying behavior is much less distinct for the test data. This can be explained by all instabilities being contained by the training set."
- "The spectra of the standardized residuals for the training and test sets can be seen to differ, as for the time series. Because the instabilities are only present in the training

data, the response characteristics of the two sets are different, which limits the validity of cross-validation. In unison with the other whiteness measures, the number of sign changes in the residual sequences for the training and test datasets is are not within the 95 % confidence intervals of the number of sign changes for an the corresponding ideal white noise sequences. This supporting supports the interpretation that the model residuals do not resemble white noise."

**Other minor edits**

- Author affiliations: "Lars Morten Sørensen42" and "Jon Juel Thomsen42"
- Throughout manuscript: Corrected spelling mistakes and made slight language changes to increase readability.
- Reference (Ebbehøj et al., 2023): "Short-term damping estimation for time-varying vibrating structures in nonstationary operating conditions, (submitted for journal publication), preprint: http://ssrn.com/abstract=4452026, Mechanical Systems and Signal Processing, 205, 110 851, 605,10.1016/j.ymssp.2023.110851, 2023."
- Changed notation of standardized residuals in captions to Figs. 6 and 11 align with Eq. (15), and emphasize it is in discrete time:  $\frac{z(t)}{z_t}$
- Fig. 9-11: Changed axis labels: "Normaliszed frequency"
- Sect. 4.3, 1st paragraph: "The estimated normalized frequencies correspond well with peaks in magnitude seen in the spectrum and spectrogram of response y in Fig. 9, and the average 95 % confidence intervals for the first and second mode are ±(0.01,0.02) Hz."